# Ensemble flood forecasting considering dominant runoff processes: I. Setup and application to nested basins (Emme, Switzerland)

Manuel Antonetti[1,2], Christoph Horat[1,3], Ioannis V Sideris[4], and Massimiliano Zappa[1]

[1]Swiss Federal Institute for Forest, Snow and Landscape Research, Birmensdorf, Switzerland
[2]University of Zurich, Department of Geography, Zurich, Switzerland
[3]ETH, Institute for Atmospheric and Climate Science, Zurich, Switzerland
[4]MeteoSwiss, Swiss Federal Office of Meteorology and Climatology, Locarno, Switzerland

*Correspondence to:* Massimiliano Zappa (massimiliano.zappa@wsl.ch)

**Abstract.** Flash floods evolve rapidly during and after heavy precipitation events and represent a potential risk for society. To predict the timing and magnitude of a peak runoff, it is common to couple meteorological and hydrological models in a forecasting chain. However, hydrological models rely on strong simplifying assumptions and hence need to be calibrated. This makes their application difficult in catchments where no direct observation of runoff is available.

To address this gap, a flash-flood forecasting chain is presented based on: (i) a nowcasting product which combines radar and rain gauge rainfall data (CombiPrecip), (ii) meteorological data from state-of-the-art numerical weather prediction models (COSMO-1, COSMO-E), (iii) operationally available soil moisture estimations from the PREVAH hydrological model, and (iv) a process-based runoff generation module with no need for calibration (RGM-PRO). This last component uses information on the spatial distribution of dominant runoff processes from the so-called maps of runoff types, which can be derived with different mapping approaches with increasing involvement of expert knowledge. RGM-PRO is event-based and parametrised a priori based on the results of sprinkling experiments.

This prediction chain has been evaluated using data from April to September 2016 in the Emme catchment, a medium-size flash-flood prone basin in the Swiss Prealps. Two novel forecasting chains were set up with two different maps of runoff types, which allowed sensitivity of the forecast performance on the mapping approaches to be analysed. Furthermore, special emphasis was placed on the predictive power of the new forecasting chains in nested subcatchments when compared with a prediction chain including a original version of the runoff generation module of PREVAH calibrated for one event.

Results showed a low sensitivity of the predictive power on the amount of expert knowledge included for the mapping approach. The forecasting chain including a map of runoff types with high involvement of expert knowledge did not guarantee more skill. In the larger basins of the Emme region, process-based forecasting chains revealed comparable skill as a prediction system including a conventional hydrological model. In the small nested subcatchments, although the process-based forecasting chains outperformed the original runoff generation module, no forecasting chain showed satisfying skill in the sense that it could be useful for decision makers.

Despite the short period available for evaluation, preliminary outcomes of this study show that operational flash-flood predictions in ungauged basins can benefit from the use of information on runoff processes, as no long-term runoff measurements are needed for calibration.

# 1   Introduction

Flash floods (FFs) arising from the interaction of the atmospheric and the hydrological system are characterised by a runoff peak that develops within time periods that range from minutes to hours and may occur during or after intense rainfall (Norbiato et al., 2008). They may result in threatening catastrophes and pose a risk to society, especially on small scale catchments (of few hundred square kilometres of size or less) with steep slopes and shallow soils. Since small basins react quickly to precipitation there is only little time for warnings (Liechti et al., 2013). Furthermore, FFs can be accompanied by landslides and mud flows (Collier, 2007). Impermeable surfaces and saturated soils may accelerate the rainfall-runoff transition (Norbiato et al., 2008).

FFs are considered to be significant natural hazards and they are associated with serious risk to life and destruction of buildings and infrastructure (Collier, 2007; Norbiato et al., 2008; Gaume et al., 2009). In Europe, FF occurrence peaks during autumn in Mediterranean and Alpine-Mediterranean areas and during summer in inland continental regions due to pronounced convective activity (Norbiato et al., 2008; Marchi et al., 2010). The magnitude of the events is in general larger in Mediterranean countries than in inner continental countries (Gaume et al., 2009; Javelle et al., 2010). According to Gaume et al. (2009), the most severe FF events in Europe were the Barcelona flood in Spain (1962) with over 400 casualties (Lopez Bustos, 1964), the two floods in the region of Piedmont in Italy (1968 and 1994) with respectively 72 and 69 victims (Ferro, 2005; Guzzetti et al., 2005) and the Aude flood in France (1999) with 35 fatalities (Gaume et al., 2004). Economic damages associated with such floods were substantial, e.g. 3.3 billion euros for the Aude flood (Lefrou et al., 2000) and 1.2 billion euros for the Garde flood which occured 2002 in France (Huet et al., 2003; Delrieu et al., 2005; Braud et al., 2010). In Switzerland, in June 2007, heavy precipitation caused flooding by the river Langeten and landslides in the region of Huttwil, Canton of Bern. This led to three fatalities and damages of 60 million Swiss francs (Liechti, 2008). In July 2014, flooding by the river Emme and landslides were responsible for damages of 15 million Swiss francs in Schangnau, Canton of Bern (Andres et al., 2015).

## 1.1   Current approaches for flash flood prediction

As both meteorological and hydrological conditions are important for FF prediction, coupled approaches were developed, as, for instance, the so-called Flash Flood Guidance (FFG) concept, which is used to issue warnings in the U.S. (Carpenter et al., 1999; Norbiato et al., 2008). FFG is defined as the rainfall depth which is necessary to occur for a certain duration to cause minor flooding in a specific basin. According to Georgakakos (2006) and Norbiato et al. (2008), for the U.S., FFG thresholds are computed with a hydrological model that is run iteratively with increasing amounts of rainfall of a given duration. The FFG provides a value of susceptibility of a basin to a FF and takes the hydrological state of the system and in particular soil moisture into account. In operational mode, FFG is computed each day. When nowcast or forecast rainfall depth is higher than FFG, a warning is issued as a flooding is likely. Although this concept is useful, neither the timing nor the magnitude of the event is assessed (Norbiato et al., 2008). As a further approach, Collier and Fox (2003) proposed a Flash Flood Susceptibility Assessment Procedure (FFSAP) which is similar to what Mani et al. (2012) elaborated for the catchments investigated here (see Sect. 2.1) and to what is currently deployed in Saxony (Eastern Germany) for operational flash-flood early warning (Philipp et al., 2016). Mani et al. (2012) developed an approach for the Swiss Emme basin based on the concept of "disposition", defined as the

susceptibility of a region to flash floods and debris flow. In their approach, the actual disposition is defined by the sum of *base* and *variable* disposition, where the former is inferred from geological properties of the catchment and the latter is dependent on time. Whether a process initiation through heavy precipitation is expected — meaning that the actual disposition reaches a threshold — is determined with analyses of rainfall radar data (Panziera et al., 2016). Although this approach provides the geographical distributions of event-prone areas, it is expensive as it requires periodic field work to sample the variable disposition. In addition, as with the FFG concept, it does not provide detailed information on the magnitude and timing of an event. Several combinations of meteorological and hydrological models were implemented in so-called forecasting chains to quantitatively predict peak flows. It was already examined by e.g. Georgakakos (1986), who implemented a stochastic-dynamic hydrometeorological model. In general, a forecasting chain consists of (a) an atmospheric model, (b) a hydrological prediction system, (c) a nowcasting tool for initial conditions and (d) warnings for end-users (Zappa et al., 2008, 2011). The advantage of this approach is that timing and magnitude of the event can be predicted. Some examples of forecasting chains are described below, with a particular focus on the hydrological model.

Rossa et al. (2010) carried out a case study for the 26 September 2007 Venice FF in the 90 km$^2$ Dese river basin. They implemented a forecasting chain with a semi-distributed hydrological-hydraulic model that is based on the Green-Ampt approach (Heber Green and Ampt, 1911) for infiltration-excess and saturation-excess runoff generation and the Penman-Monteith equation (Penman, 1948; Monteith et al., 1965) for evapo-transpiration fluxes. As the river network of their study area is affected by tides, the coupling to the hydrodynamic model was of importance. Liechti et al. (2013) explored the potential of two radar-based ensemble forecasting chains for FF early warning in alpine catchments in southern Switzerland, including the Verzasca basin. They found that it is valuable having an ensemble in hydrological initial conditions. However, data needed for such predictions are only available in certain regions as they are exclusively produced for research projects, which prevents the operational application of their approach. A skilful forecasting chain for the river Sihl and the city of Zurich was developed by Addor et al. (2011), which combined PREVAH hydrological model with FLORIS hydraulic model and used deterministic and probabilistic meteorological input. In all of the forecasting chains of Liechti et al. (2013) and Addor et al. (2011), the hydrological model used (PREVAH, Viviroli et al. (2009)) relied on calibration.

Haag et al. (2016) integrated spatially distributed information on dominant runoff processes (DRPs, see Sect. 3.1.1) based on the classification of Scherrer and Naef (2003) into LARSIM (Large Area Runoff Simulation Model, Bremicker (2000)). For each DRP, the soil module was parameterised based on numerical experiments. Depending on which runoff process is dominant in a certain Hydrological Response Unit (HRU), a corresponding basis parametrisation of the soil storage was used. With scaling factors, which allow for an adaptation of the basis parametrisation, the model was calibrated while maintaining geographical heterogeneity. Subsequently, LARSIM was forced with meteorological input data and applied on the Nahe catchment in Rheinland-Pfalz, where it is used for operational flood prediction and early warning since 2014.

## 1.2  Challenges and uncertainties

FF predictions are challenging for several reasons. A first challenge is that FFs are extreme events and often occur in ungauged basins, which means that there is only little data available for their investigation (Gaume et al., 2009). In most extreme cases, hydrometric measuring devices may even be destroyed (Collier, 2007). In order to still have sufficient data to perform a proper statistical analyses, warning thresholds are often set too low and become not relevant for FFs anymore (Liechti et al., 2013). Furthermore, especially small catchments are prone to FFs (Alfieri et al., 2011), which requires a high resolution of the forcing numerical prediction model (Collier, 2007).

Considering a forecasting chain, uncertainties of meteorological input, of the hydrological initial conditions, of the structure of the hydrological model and of the hydrological model parameters propagate and superpose through the flood forecasting chain in a non-linear fashion (Velazquez et al., 2011; Zappa et al., 2011). *Meteorological uncertainty*, which can be assessed with a meteorological ensemble prediction system, is usually assumed to account for the largest share of total uncertainty (Rossa et al., 2011; Zappa et al., 2011). *Uncertainty in the parameters of the hydrological model* follows from an incomplete understanding on how to mathematically represent the rainfall-runoff transition process and can be treated with a hydrological multi-model approach (Fenicia et al., 2011; Velazquez et al., 2011). *Hydrological model parameter uncertainties* result when physical processes affecting runoff generation are modelled conceptually and multiple parameter sets are identified during the calibration process that lead to optimum model performance, a problem which is known as equifinality (Beven, 1993). Zappa et al. (2011) treated uncertainty in model parameters with an ensemble of the PREVAH hydrological model and found this uncertainty source being responsible for the second largest contribution to the total uncertainty in their study.

## 1.3  Prior work and objectives

From the previous literature review, it emerges that in recent years several studies on the topic of FFs prediction have been completed by our research group. Before declaring the goals of this new paper it is useful to have a summary of our prior work on related topics. Table 1 presents an overview on seven papers elaborated since 2011. Zappa et al. (2011) is our benchmark paper on uncertainty propagation and evaluated probabilistic forecasts in the Verzasca basin, for which real-time forecasts as forced by different generation of forcing numerical weather prediction models (NWP: COSMO-1, COSMO-2, COSMO-LEPS and COSMO-E on Table 1) have been operated since 2007 (Zappa et al., 2008). Addor et al. (2011) is the first work where we address the topic of verification of deterministic and ensemble forecasts and focussed on the river Sihl. Liechti et al. (2013) investigated flash-flood nowcasting with advanced ensemble weather radar products and a deterministic NWP for three areas in southern Switzerland, including the Verzasca river. Antonetti et al. (2017) first introduces RGM-PRO and its a-priori configuration. RGM-PRO is a runoff generation module (RGM) with no need for calibration. Re-simulating sprinkler experiments (Kienzler and Naef, 2008) allows an a priori determinination of the parameters generally requiring calibration (see Sect. 3). This procedure has been evaluated in five target areas, including a sub-basin of the Emme catchment. Antonetti and Zappa (2018) investigated with different configurations of RGM-PRO to which extent expert knowledge can improve simulation results under consideration of uncertainty in the Emme catchment and its main tributaries. All the studies confirmed

that forecasts of timing and magnitude of flash-floods are of importance, and require combination of a meteorological prediction with a hydrological model. The latter could either be a physically-based model — which is computationally expensive and not the first choice for operational use — or a conceptual model, where calibration problems arise and hinder applicability for ungauged basins. Larger catchments and low-flow periods are predicted by current state-of-the art forecasting chains relying on calibration (Zappa et al., 2008). This study and the companion paper by Horat et al. (2018) have been designed to evaluate possible operational deployment of an event-based runoff generation module such as RGM-PRO, which has the potential of being configured for ungauged areas without needs for tailored calibration. Such event-based tools should be only operate when thunderstorms are to be expected and provide information to anticipate flash-floods in small fast-reacting areas. These two latest experiments are the firsts using the COSMO-E and COSMO-1 numerical models for hydrological predictions (see section 2.2). The period of evaluation covers the summer of 2016. It is a short period for a comprehensive assessment of RGM-PRO, but it might already provide indication of its potential. The first research question of the present study addresses this aspect:

> To what extent does the skill of the FF prediction depend on the use of model structures considering spatially distributed information on runoff processes into a hydrological model?

The new RGM which includes knowledge on runoff processes is expected to be advantageous over traditional RGMs in nested subcatchments (Antonetti et al., 2016a). The reason for this is that the calibration procedure for the common hydrological model is performed with data from the runoff gauge of the main catchment. Therefore, the second research question is the following:

> Is it possible to increase skill in FF forecasting in nested subcatchments with the use of a process-based RGM which includes spatially distributed information on DRPs instead of using a traditional RGM relying on calibration in a forecasting chain?

A forecasting chain with state-of-the-art meteorological and hydrological components is proposed and evaluated for the Swiss Emme catchment (see Sect. 2.1). The different components of the forecasting chains are described in Sect. 3. In Sect. 3.3, the skill assessment procedure used for this study is presented. To avoid large overlaps in the presentations of the methods, the methods section of the present paper focusses on the hydrological component of the FF forecasting chain, whereas the companion paper (Horat et al., 2018) presents more information on the used NWP COSMO-E and COSMO-1 and applies RGM-PRO in forecasting mode for the Verzasca catchment and compare its quality with our current operational model. The results are shown in Sect. 4 and are discussed in Sect. 5. In Sect. 6, the conclusions are drawn.

## 2 Target area and data

### 2.1 Target area

The Emme catchment (445 km$^2$), shown in Fig.1, is located in the Prealps and lies mainly in the Canton of Berne. It ranges from 560 to 2120 m a.s.l.. The Trueb subcatchment (55 km$^2$) is nested within the Ilfis subcatchment (184 km$^2$), which is in turn nested within the main catchment, here also referred to as Emmenmatt. Another nested subcatchment of the main catchment is Eggiwil, which is 125 km$^2$ in area. Considering land use, 4 % of the basin is covered by settlements, 52 % by pasture and 44 % is forested. In the catchment, geological sequence of Upper Freshwater Molasse, Upper Marine Molasse, Lower Freshwater Molasse, Flysch and limestone is present. For a more detailed description of the study area we refer to Antonetti and Zappa (2018).

Runoff measurements at hourly resolution for comparison with simulations were provided by the Swiss Federal Office for Environment (FOEN) for the Emmenmatt, Eggiwil and Ilfis catchments. For the Trueb catchment, measurements from the Bau-, Verkehr- und Energiedirektion of the Canton of Berne were available. For the evaluation of hindcasts, only four events are investigated as runoff data is not available from 2005 to 2010 for the Trueb catchment.

### 2.2 Meteorological forcing

Accurate precipitation estimation is a demanding task. Rain gauges provide relatively accurate precipitation measurements on the ground, but their spatial representativeness may be low depending on the aggregation used and the type of weather. Moreover, rain gauge networks can cover only sparsely large regions, therefore important features of the precipitation field may be missed, since rainfall is typically characterised by high spatiotemporal variability (Liechti et al., 2013; Sideris et al., 2014). Radar precipitation estimates cover large regions densely but these estimates are essentially mean values for grid cells with resolution of 1x1 km$^2$. Moreover, such estimates are subject to complex errors generated through the chain of processes of signal transmission, hydrometeors backscattered signal detection, and their eventual transformation into units that characterise precipitation water(Germann et al., 2006).

Combination of radar and rain gauge precipitation estimates (CombiPrecip) is in essence a localized adjustment of the radar field using the rain gauge measurements. Such techniques typically involve geostatistics and produce as an output an optimal field which is unbiased in comparison with the existing rain gauge measurements. It also maintains the underlying spatial structure associated with the radar rainfall observation. CombiPrecip is a techique developed by Swiss Federal Office of Meteorology and Climatology MeteoSwiss which relies on spatiotemporal geostatistics to produce this adjustment (Sideris et al., 2014). Information on hourly rainfall at ground stations is blended to the weather radar signal. With respect to the target area, the most representative rain station used in the blending is the one on the "Napf" (Fig.1). CombiPrecip has been useed in this study in order to force the event-based runoff generation modules with gridded precipitation for the selected events. Retrospective CombiPrecip data are available for the period 2005 to 2013. The data used here have been collected and archived during real-time operations of systems developed by WSL.

As future rainfall input, quantitative precipitation forecasts were used from NWP by MeteoSwiss, namely COSMO-E and

COSMO-1, and were processed as in Addor et al. (2011). COSMO-1 has a grid spacing of 1.1 km and runs as deterministic model with initialisations every three hours. Lead time is 33 hours except for the 03 UTC run, where a 45 hour forecast is available. COSMO-E is an ensemble prediction system with 2.2 km grid spacing, two initialisations each day and a lead time of 120 hours. Both COSMO-E and COSMO-1 are available for only one season and there is no prior experience in applying these

models in a forecasting chain. A recent evaluation of COSMO-E is presented in Klasa et al. (inpress), while its configuration can be found accessing the MeteoSwiss webpage (MeteoSwiss, 2018).

## 3   Models and Methods

### 3.1   Process-based modelling with RGM-PRO

#### 3.1.1   Mapping dominant runoff processes

Information on the spatial distribution of runoff processes in a catchment can be visualised in maps of RTs (e.g. Schmocker-Fackel et al. (2007)). Such maps are necessary for RGM-PRO simulations and can be generated in various ways. For identification of the DRPs at the plot scale, Scherrer and Naef (2003) developed a decision scheme based on the possible flow paths of water on temperate grassland. In each vertical soil compartment — from the surface to underlying geology — the flow process is determined by critical factors. These can be for instance the vegetation cover for the surface compartment, macropores for

the topsoil, lateral preferential pathways for the subsoil and permeability for the geological underground. At the end of each possible path through the system, the occurring DRP is identified. For other land use types such as arable land or forest, different decision schemes are used. In order to upscale the DRPs from the plot to the catchment scale, Schmocker-Fackel et al. (2007) developed a simplified method relying on a soil map and a high resolution digital terrain model (2x2 meters) in a GIS environment. During the upscaling, the DRPs are reclassified into RTs according to Table 2 (Antonetti et al., 2016a, 2017). In

regions where soil data is absent, a relatively time-consuming soil model based on expert knowledge was used (Margreth et al., 2010). Maps of RTs generated with this methodology are referred to as *Margreth map* and were already used, for instance, in Antonetti et al. (2016a, 2017). A simpler upscaling approach with less involvement of expert knowledge was proposed by Müller et al. (2009), and the resulting maps are onwards referred to as *Müller map*. The method of Müller et al. (2009) relies on a digital terrain model at lower resolution than *Margreth map* (25x25 meters in this application), a geological map, and a land

use map. It assumes that mainly slope and permeability of the geological substratum determine DRPs, whereby information about soil characteristics are not needed. Geological substratum is classified into permeable and impermeable, land use data into grass-, arable land and forest, and slope into five categories. For any land use class and slope, regions with permeable substratum are classified as deep percolation, i.e. RT5. DRPs for impermeable geology depend on a combination of land use and slope category (Müller et al., 2009).

For the Emme region, a Margreth map from SoilCom GmbH was available and used for this study (Fig. 1, left). In addition, a map of RTs with the approach of Müller et al. (2009) was derived (Fig. 1, right).

Antonetti et al. (2016a) compared the similarity of various DRP mapping approaches including high to low amount of expert

knowledge in the Meilen and the Reppisch catchments on the Swiss Plateau. A manually derived map after Scherrer and Naef (2003) served as reference. Furthermore, they performed synthetic runoff simulations in order to assess the sensitivity of the hydrograph on the mapping approach. Antonetti et al. (2016a) found that the simulations with the most complex automatic mapping approach, i.e. Schmocker-Fackel et al. (2007), produced the most similar results when compared with the reference. In contrast, simulations following the simplified schemes lead to the strongest deviations.

Here we evaluate the role of maps in another region and focus on the role of mapping approaches on flood predictions in operational setup.

### 3.1.2 Structure and a priori parametrisations

RGM-PRO is a standalone runoff generation module and in many terms a spin-off of the traditional module of the PREVAH hydrological model (Viviroli et al., 2009). It integrates knowledge on runoff generation mechanisms as elaborated in Antonetti and Zappa (2018). A comparison of the traditional structure of the PREVAH module (RGM-TRD) with RGM-PRO is shown in Fig. 2. RGM-PRO uses information contained in maps of RTs (Fig. 2, b), which are based on the classification of DRPs (Table 2, Scherrer and Naef (2003)). The parameter values of RGM-PRO are determined a priori for each RT by re-simulating sprinkler experiments (Kienzler and Naef (2008), Scherrer et al. (2007), Fig. 2, a). With this approach, RGM-PRO can avoid classic calibration against runoff observations which allows the application in areas where no measurements are available. Therefore, it is more advantageous having five parameters that can be obtained a priori than one parameter for which calibration is needed. The meaning of the RGM-PRO parameters in Fig. 2 is presented in Table 3. For a detailed description of the model please refer to Antonetti et al. (2017).

RGM-PRO runs at a spatial resolution of 500 m and a temporal resolution of one hour and requires gridded precipitation input. Maps of RT are available at higher spatial resolution (25 by 25 m$^2$) and used to obtain a sub-grid parametrisation of RGM-PRO. Specifically, for each 500x500 m$^2$ cell the percentage of each RT is taken into account (Antonetti et al., 2017). As RGM-PRO is an event-based model and soil moisture data is needed for initialisation. For this, an operationally available gridded input from PREVAH at a spatial resolution of 500x500 m$^2$ was used. This PREVAH version coincided with the one used for real-time assessment of drought in Switzerland (Zappa et al., 2014; Jörg-Hess et al., 2015). Subsequently, soil moisture data was downscaled to a resolution of 25 m according to Blöschl et al. (2009). With this method, the map of RTs serves as fingerprint since it contains information determining the spatial variability of soil moisture (Antonetti and Zappa, 2018).

### 3.1.3 Traditional benchmark version with conventional hydrological runoff generation module

For comparison, a forecasting chain with the traditional structure (RGM-TRD) was set up. Ideally RGM-TRD needs to be calibrated and validated against several observed flood events. In this application we decided to calibrate on one single event, the largest runoff event measured at Emmenmatt gauge in 2016 which occurred on the $14^{th}$ of May. We choose this approach in order to evaluate a setup with minimum requirements concerning observed discharge. This should show the potential of the RGM-TRD approach, when a single measurement campaign is available, as discussed for example in Pool et al. (2017).The

calibrated parameter set, valid for the whole target area, finally resulted from the ten runs with highest Kling-Gupta efficiency (KGE, see Sect. 3.3) out of 4'000 Monte Carlo simulations, as conducted also for the study by Antonetti and Zappa (2018).

## 3.2 Overview of completed experiments

Experimental setup for the Emme areas is displayed in Fig. 3. Two forecasting chains with RGM-PRO were set up based on
Müller map (DRP-mu-C1 and DRP-mu-CE) and two based on Margreth map (DRP-ma-C1 and DRP-ma-CE). Comparison of these will show possible advantages of including high amount of expert knowledge in the map of RTs for forecasting purposes. In addition, two chains were built with the calibrated RGM-TRD, namely PRE-C-C1 and PRE-C-CE. Comparison of RGM-PRO-based chains with the ones based on the calibrated PREVAH indicates whether a hydrological model integrating knowledge on DRPs can compete with a calibrated one in forecast mode. The outcomes of the Monte Carlo experiment have
been used to evaluate RGM-PRO and RGM-TRD for past events in the Emme area. All forecasting chains relied on model initialisation with CombiPrecip and soil moisture data from PREVAH simulations. Furthermore, for all forecasting chains, start of the simulation was at the moment in time with minimum observed runoff in the last five days prior to the forecast.

Our investigation period was restricted to May until July 2016. The reason for this is that the NWPs with the new COSMO-E and COSMO-1 models are only available since March 2016. The prior models COSMO-LEPS and COSMO-2 (e.g. Zappa et al.
(2008)) have been dismissed in September 2016 and will not be available for future operational deployment. A comparison between the prior and new NWP models is outside the scope of this study.

The strength of the approach presented in this study including the new RGM is that it does not require calibration. Although it is of advantage having an ensemble in hydrological initial conditions (Liechti et al., 2013), the use of probabilistic nowcasting tools to treat this kind of uncertainty is renounced. This seems justified as spread decays within the first 48 hours (Zappa et al.,
2011) and this study aims to have a system in operational mode. Furthermore, uncertainty due to the hydrological model structure is not addressed here. Studies on this topic have been presented by Seiller et al. (2017) and Perrin et al. (2001).

## 3.3 Data analysis

For the verification of deterministic forecasts for continuous predictands, the Nash–Sutcliffe efficiency (NSE) was computed (Nash and Sutcliffe, 1970). As a precursor of NSE, the mean squared error (MSE, Eq. (1)) was calculated as the arithmetic
average of the squared difference of the forecast-observation pairs.

$$MSE = \frac{1}{n} \sum_{k=1}^{n} (y_k - o_k)^2 \qquad (1)$$

The NSE is then obtained with building the skill score of MSE, given in Eq. (2). A skill score describes the improvement of a specific forecast over a reference forecast, where the mean runoff during the events served as reference forecast, shown in Eq. (3).

$$NSE = 1 - \frac{MSE}{MSE_{clim}} \tag{2}$$

with

$$MSE_{clim} = \frac{1}{n} \sum_{k=1}^{n} (\bar{o} - o_k)^2 \tag{3}$$

In addition, with Eq. (4) the Kling–Gupta efficiency (KGE) was computed (Gupta et al. (2009)), which is a decomposition of NSE into a linear correlation ($r$), a bias ($\beta$) and a variability of flow component ($\alpha$).

$$KGE = 1 - \sqrt{(r-1)^2 + (\alpha-1)^2 + (\beta-1)^2} \tag{4}$$

The perfect value of NSE and KGE is one and positive values indicate benefit compared with a reference forecast.

For deterministic and probabilistic forecasts, the Brier score (BS, Eq. 6) and the Brier skill score (BSS, Eq. 5) were computed. As these scores evaluate dichotomous predictands, a quantile of hourly runoff climatology from May to July 2016 served to distinguish between event and non-event. A value of zero for BS and a value of one for BSS is achieved by a perfect forecast. For BSS, the mean runoff from May to July 2016 served as reference forecast (Eq. 7):

$$BSS = 1 - \frac{BS}{BS_{clim}} \tag{5}$$

with

$$BS = \frac{1}{n} \sum_{k=1}^{n} (y_k - o_k)^2 \tag{6}$$

and

$$BS_{clim} = \frac{1}{n} \sum_{k=1}^{n} (\bar{o} - o_k)^2 \tag{7}$$

For the ensemble predictions, area under Receiver Operating Characteristics (ROCa) was computed, which is a well suited measure to establish a synthesis across methods and lead times. Furthermore, with ROCa, the utility of a prediction system for end-users can be assessed. According to Buizza et al. (1999), a ROCa value of 0.7 is the minimum value useful for decision

makers. ROCa is used here to summarise the outcomes presented also in the supplementary information. A matrix is compiled to visualise which of the used chains is performing best. In general, the verification of hydrological ensemble predictions accounts for the recommendations issued by Brown et al. (2010) and used in previous applications (Addor et al., 2011; Liechti et al., 2013).

## 4 Results

### 4.1 Evaluation of hindcasts

For eight large runoff events from 2005 to 2016 and using the configuration and evaluation strategy described in Antonetti et al. (2017), we compared the performance of uncalibrated (TRD-NC) and calibrated (TRD-C) PREVAH and RGM-PRO with Margreth (DRP-ma) and Müller map (DRP-mu) when forced exclusively with CombiPrecip data (Fig. 4). A calibration was completed as stated in Section 3.1.3 for the Emmenmatt gauge, the results for the nested Ilfis, Eggiwil and Trueb catchments represent an internal verification.

For a comparison of the mapping approaches for RGM-PRO, the coloured border in Fig. 4 defines whether DRP-ma (pink) or DRP-mu (light green) performs better in terms of median KGE. It reveals that DRP-ma outperforms DRP-mu in 18 out of 28 cases. However, when not considering the Eggiwil catchment, the two models perform best exactly an equal number of times, i.e. 10 out of 20 cases.

Considering uncalibrated and calibrated PREVAH as well demonstrates that TRD-NC is worst in terms of KGE in by far the most cases. In addition, uncertainty is always largest. Comparing calibrated PREVAH with RGM-PRO approaches indicates that TRD-C is better than both DRP-ma and DRP-mu in only 5 out of 28 times. However, performance of the three latter mentioned models is comparable and highly depending on the event-catchment combination. In Trueb basin, DRP-mu seems to be preferred over DRP-ma and the DRP-based approaches over the calibrated PREVAH. However, there are only four events as runoff data is not available from 2005 to 2010.

### 4.2 Preliminary assessment of re-forecasts

#### 4.2.1 Example of re-forecasts

Visual analysis of events remains a very valuable complement to statistical evaluation and gives to forecasters and users a better intelligible way to get a feeling on the quality of their systems. Fig. 5 depicts deterministic flood predictions of DRP-ma-C1, DRP-mu-C1 and PRE-C-C1 in Emmenmatt basin, as well as the temporal evolution of precipitation and soil moisture from May 11 to May 15, 2016. This was the largest event in the Emmenmatt catchment investigated in this study and also the time period for which PRE-C-C1 (and PRE-C-CE) was calibrated. In terms of COSMO-1 precipitation forecasts, cumulated predicted rainfall reveals to be in good agreement with CombiPrecip data and leads to a gradual increase in soil moisture, which started from slightly unsaturated conditions as assimilated from the real-time product introduced in Sect. 3.1.2. Main phases of precipitation input are in the late evening of May 12 and in the early morning of May 14. Corresponding peaks in observed

runoff lag only few hours behind, which is a consequence of the fast responding properties of Emmenmatt catchment. None of the three prediction chains is really able to catch the quickly rising hydrograph during initialisation period with CombiPrecip, although performance is satisfying. Simulated first peak of DRP-ma-C1 and PRE-C-C1 is relatively good in terms of volume, whereas DRP-mu-C1 underestimates it. These characteristics appear as well in forecast mode, where highest forecasts of DRP-ma-C1 and PRE-C-C1 reach up to observed second peak but are substantially below for DRP-mu-C1. Overall, performance of DRP-ma-C1 and PRE-C-C1 is comparable in this example. Comparing the two DRP-based RGMs, Margreth mapping approach reveals to be generating higher peaks than the Müller approach. This holds as well when considering more examples, however, it is not always of advantage. Spread in hydrographs resulting from the eight COSMO-1 initialisations on that day is considerable, especially for the strongly reacting DRP-ma-C1 and PRE-C-C1. This jumpiness is a typical behaviour of deterministic systems and a major problem for decision makers still relying on such systems (e.g. Bruen et al. (2010)). Probabilistic forecasts for the same event from DRP-ma-CE, DRP-mu-CE and PRE-C-CE are depicted in Fig. 6, with a switch to forecast mode on May 12, 19:00. At the start of the forecasting period, all three prediction chains are overconfident, as ensemble spread in precipitation input has not yet developed. Afterwards, observed runoff is contained within ensemble range for all forecasting chains, except for parts of the recession period. As for deterministic predictions, DRP-ma-CE and PRE-C-CE perform very similar in this example.

### 4.2.2   Evaluation of short range forecasts with hydrological measures of agreement

NSE and KGE are used to provide a classical hydrological way of evaluating the experiments. When considering NSE and KGE for Emmenmatt and Eggiwil catchments, shown in Fig. 7, it can be seen that there is skill for all deterministic forecasting systems and all investigated lead times and that skill declines over time.

In Emmenmatt catchment, DRP-ma-C1 reveals the poorest and DRP-mu-C1 the best performance in terms of NSE. Findings from KGE imply no clear preference for one forecasting chain. In Eggiwil basin, all three forecasting chains perform equally well. For the Ilfis catchment, there is skill for all forecasting chains and any lead time except for DRP-ma-C1 after six hours in terms of NSE. Values of both NSE and KGE are decreasing over time. For all lead times, DRP-mu-C1 is the best and DRP-ma-C1 the worst performing forecasting chain, apart from a lead time of 29 hours where PRE-C-C1 is best in terms of KGE. In the Trueb basin, there is no skill for any forecasting chain in terms of NSE. Considering KGE, there is little skill for DRP-mu-C1 at lead times of 12, 24 and 29 hours and for PRE-C-C1 at 24 and 29 hours lead time.

### 4.2.3   Comparison of deterministic and probabilistic forecasts

The BSS is used as proposed in Addor et al. (2011) as a measure for comparing deterministic and probabilistic forecasts, with the boxplots representing the sampling uncertainties of the score computations obtained with bootstrapping. In all catchments, there is in general a decrease of BSS with lead time, which is in particular strong for Emmenmatt (Fig. 8). Figures of BSS for Eggiwil, Ilfis and Trueb catchments are provided in the supplementary materials. Furthermore, there is less skill for increasing threshold quantiles. For deterministic forecasts there is mostly no skill for $q_{0.975}$ and $q_{0.99}$ quantiles in Emmenmatt, Eggiwil

and Ilfis, whereas in Trueb there is only skill for the $q_{0.7}$ threshold quantile.

In all basins, no increase of uncertainty with lead time is visible and spread of forecasting chains relying on CE is larger than for approaches based on C1. The ensemble approach is always better than its respective deterministic counterpart with few exceptions. This confirms numerous previous studies on the topic. Deterministic forecasting chains are most competitive at short

lead times, whereas for lead times of 24 and 29 hours skill of ensemble approach is substantially larger. One of the reasons for this behaviour could be the higher resolution of COSMO-1 as compared to COSMO-E and thus the better consideration of convection in the deterministic apporach. Nevertheless the analysed data set is too short for conclusive statements on this finding. In most cases, however, uncertainty bars of BSS overlap.

Comparing the deterministic forecasting chains based on Müller and Margreth mapping approaches in Emmenmatt reveals

that DRP-mu-C1 is better than DRP-ma-C1 for all quantiles apart from the low ones and long lead times, where BSS values are comparable. In Eggiwil and Trueb basins, performance of DRP-ma-C1 and DRP-mu-C1 is similar, with a slight preference for Margreth approach in Eggiwil. In Ilfis catchment, DRP-mu-C1 is favoured over DRP-ma-C1 for all thresholds and lead times when there is skill.

In Emmenmatt, Eggiwil and Ilfis catchments, there is no clear preference when comparing the deterministic process-based

forecasting chains with PRE-C-C1, as performance depends on threshold quantile and lead time. For the few cases when there is skill in Trueb basin, process-based forecasting chains perform better than PRE-C-C1.

Comparing the three probabilistic forecasting chains among each other leads to different outcomes in the four Emme catchments. In Emmenmatt and Ilfis, DRP-mu-CE is slightly better than DRP-ma-CE in most cases whereas in Eggiwil, the opposite is true. In Trueb catchment, no clear preference for Margreth or Müller approach is found. In Eggiwil and Ilfis, it is not ob-

vious whether to favour PRE-C-CE or the process-based forecasting chains. In the Emmenmatt basin, in general one of the two process-based forecasting chains performs better than PRE-C-CE, although this is highly depending on threshold and lead time. A clear advantage is found for process-based forecasting chains in the Trueb basin, as skill — if there is any — is usually substantially larger than for PRE-C-CE.

### 4.2.4 Synthesis of extended-range forecast quality with COSMO-E

In order to compare the different forecasting chains DRP-ma-CE, DRP-mu-CE and PRE-C-CE on their performance in terms of ROCa, a summary is shown in Fig. 9. The values of ROCa that served as a basis for Fig. 9 can be found in the supplementary materials. In no catchment there is a clear pattern or preference for one approach. Out of 87 cases, which is the number of either dark orange, pink or yellow coloured squares, DRP-ma-CE is best 38, PRE-C-CE 31 and DRP-mu-CE 18 times. For all ensemble approaches and catchments, values of ROCa are in general not higher than 0.9 and decrease with lead time.

For longest lead times, values of ROCa tend to be around 0.7, i.e. at the boundary of being useful (Buizza et al., 1999). An exception is found in the Trueb basin, where highest quantiles have largest values of ROCa (around 0.8) for longest lead times. In contrast, for the $q_{0.975}$ threshold quantile at 113 hours lead time there is clearly no skill in Emmenmatt and Eggiwil and there was not enough data for the computations to be executed in Ilfis basin.

## 5 Discussion

### 5.1 Effect of different DRP mapping approaches

In terms of most measures of agreement, no clear preference for using either the Margreth or Müller map in FF forecasting chains is found, with uncertainty bars overlapping in most cases. However, there is a modest advantage for the Müller mapping approach. For Emmenmatt, Ilfis and Trueb catchments, DRP-mu-C1/CE perform slightly better than DRP-ma-C1/CE for most lead times in terms of NSE, KGE and BSS.

The case studies with the forecasting chains reveal that DRP-ma-C1/CE react more intense on precipitation in comparison with DRP-mu-C1/CE. This does not necessarily lead to faster occurring peaks, but to peaks that are higher in magnitude. This is in good agreement with the relative shares of RTs in Table 2 for the different mapping approaches: as the Müller method classifies much more deep percolation (RT5) there is less simulated water at the runoff gauge. In terms of peak timing, there is not much difference between process-based forecasting chains, which could be because Müller classifies also a higher fraction of RT1, leading to compensation effects. Both visual inspections of hydrographs and measures of agreement reveal very similar performance of both approaches which is remarkable when taking into account how distinct the two maps in Fig. 1 look. A reason that this difference was not as large as expected could be that — in agreement with Zappa et al. (2011) — meteorological uncertainties are dominant, and uncertainty in DRP mapping is of minor importance in forecast mode. However, also re-analysis with CombiPrecip data (Fig. 4) showed no clear preference between the two process-based forecasting chains when not considering Eggiwil catchment. These results are in agreement with Antonetti and Zappa (2018), who find that satisfying model performance is also achieved using mapping approaches with low involvement of expert knowledge. Due to considerable uncertainties there is only a slight increase in potential value when a more complex map of RTs is considered, even when using optimal runoff and real-time meteorological input data (Antonetti and Zappa, 2018).

In the Eggiwil basin, there are ongoing discussions whether runoff gauge overestimates runoff (Antonetti and Zappa, 2018). This is supported by the fact that all three forecasting chains reveal strong under-forecasting for the highest threshold quantiles, which is in contrast to the other catchments. However, it could also be possible that none of the three models is able to represent the processes important for runoff formation correctly in Eggiwil. As Margreth approach generates usually highest runoffs, it is most competitive in Eggiwil basin.

Overall, the analysis shows that, in forecast mode, a map of RTs with high involvement of expert knowledge does not guarantee a higher performance compared with a map of RTs based on less expertise. The Müller mapping approach is faster to implement, financially more attractive, and does not lead to worse results compared with the Margreth method. This finding is in agreement with Antonetti and Zappa (2018).

### 5.2 Effect of integrating knowledge on DRP into hydrological modelling in nested catchments of Emme region

Comparing the two process-based forecasting chains with the one including a calibrated hydrological model indicates comparable performance in terms of all measures of agreement used in Emmenmatt, Eggiwil and Ilfis catchment. If there is skill in Trueb basin in terms of BSS and KGE, which is hardly ever the case, performance of process-based forecasts is substantially

better than the one of PRE-C-C1/CE. This is in accordance with Antonetti et al. (2016a) and Antonetti and Zappa (2018), who state that process-based forecasting chains should be advantageous especially in nested subcatchments and not in the main catchments where the calibration was made for. However, as with the discharge measurement in Eggiwil, Scherrer AG (2012) doubts the quality of the runoff data for the Trueb basin and Antonetti and Zappa (2018) explain the poor model performance in this area to be a consequence thereof.

## 6   Conclusions

This study investigated the potential of a process-based runoff generation module for flash flood prediction from May to July 2016 in the Emme catchments. The main task was to set up four quasi-operational forecasting chains for Emmenmatt, Eggiwil, Ilfis and Trueb basin. The innovation in the approach followed in this paper is the use of RGM-PRO, a conceptual hydrological module with no need for calibration (Antonetti et al., 2017). RGM-PRO relies on spatially distributed information of runoff processes in a catchment, i.e. the so-called maps of runoff types. For all forecasting chains, either numerical weather prediction data from deterministic COSMO-1 or probabilistic COSMO-E served as meteorological input. To initialise the hydrological model, gridded precipitation nowcasts from CombiPrecip were used, which also provided meteorological input for the corresponding reference runs. The forecasting chains including RGM-PRO were set up each depending on maps of runoff types with little (Müller) and high amount of expert knowledge (Margreth), resulting in the forecasting chains DRP-mu-C1/DRP-mu-CE and DRP-ma-C1/DRP-ma-CE. This design allowed for a sensitivity analysis of the forecast performance on spatially distributed information of runoff processes. In addition, two forecasting chains including a conventional hydrological module relying on calibration (PREVAH, Viviroli et al. (2009)) were set up (PRE-C-C1/PRE-C-CE). Special emphasis was placed on the performance of the prediction systems in nested subcatchments.

> Results suggest that DRP-mu-C1/DRP-mu-CE have slight advantages in terms of most measures of agreement and catchments when compared with forecasting chains involving high knowledge for maps of runoff types. The faster implementation speed of the Muller mapping approach is an additional reason for considering it as our preferable choice in a forecast mode.

> Forecasting chains with integrated knowledge on runoff processes revealed comparable performance as the ones employing a conventional hydrological model in the larger catchments of the Emme region. In the smallest nested subcatchment, i.e. Trueb basin, prediction systems based on RGM-PRO outperformed the ones relying on conventional PREVAH substantially in the few cases when there was any skill. This confirms the potential of including information on dominant runoff process in hydrological models, as an a priori parametrised (i.e. non calibrated) hydrological model reached comparable – if not even better – results than a conventional hydrological model.

For a future study, it would be informative to assess sensitivity of spatial distribution of rainfall, i.e. whether precipitation falls on a fast or a slow reacting surface, on flood response. For this, numerical experiments with simulated thunderstorms of same intensity and duration but at different locations in a catchment could be executed in a similar way as in Paschalis et al.

(2014) and Lumassegger et al. (2016), who investigated streamflow response on space-time variability of precipitation. To quantify uncertainty in runoff type mapping approaches and propagation through the forecasting chain, an extension of Zappa et al. (2011) and Antonetti and Zappa (2018) could be possible.

We can conclude that a skilful application of a hydrological model with no need for calibration in a forecasting chain is possible, which extends the prediction of flash floods to ungauged and nested basins. In the companion paper Horat et al. (2018), the application of RGM-PRO is extended to another climatic region of Switzerland (the Verzasca basin in Southern Switzerland) and tested against the already operational forecasting chain relying on the PREVAH model (Zappa et al., 2011).

## 7  Data availability

The Margreth map used for this study is provided by SoilCom GmbH (http://www.soilcom.ch). For the Müller map, Arealstatistik 1979/85 (http://www.bfs.admin.ch) with 100 m resolution serves as land use map and DTM25 (Data: BFS GEO-STAT/Federal Office of Topography swisstopo, http://www.swisstopo.admin.ch) with a resolution of 25 m as digital terrain model. As a geological map, Geologischer Atlas GA25 (Data: BFS GEOSTAT/Federal Office of Topography swisstopo) with a scale of 1:25'000 is used where it is available and elsewhere Geologische Karte (Data: BFS GEOSTAT/Federal Office of Topography swisstopo) with a scale of 1:500'000. Meteorological data, i.e. CombiPrecip, COSMO and rain gauge data, is provided by the Swiss Federal Office of Meteorology and Climatology (MeteoSwiss, http://www.meteoswiss.admin.ch). Runoff measurements are obtained from Swiss Federal Office for Environment (FOEN, http://www.bafu.admin.ch) and Bau-, Verkehr- und Energiedirektion of the Canton of Berne (http://www.bve.be.ch).

*Author contributions.*  The study design was in the responsibility of Massimiliano Zappa and Manuel Antonetti. Massimiliano Zappa prepared the COSMO data for the simulations performed by Manuel Antonetti and Christoph Horat. The statistical analysis was carried out by Christoph Horat with the aid of Manuel Antonetti. In addition, Manuel Antonetti and Christoph Horat prepared the manuscript, where Ioannis Sideris created and provided the CombiPrecip product used. The manuscript benefited from the contributions of all co-authors.

*Acknowledgements.*  This work was carried out at Swiss Federal Institute for Forest, Snow and Landscape Research (WSL), Birmensdorf, Switzerland. The contribution of Manuel Antonetti has been funded by the Swiss Federal Office for Environment (FOEN). We would like to thank Simon Scherrer from Scherrer Hydrologie AG and Michael Margreth from SoilCom GmbH for the maps of runoff types. In addition, we are grateful to Prof. Heini Wernli (ETH Zurich) for his valuable comments and to Katharina Liechti for providing R scripts used in the verification of the forecasts.

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

**Table 1.** Overview of previous own work on the manuscript topics. Prior and current applications are classified according to the target area, the used models, the avaialble forcing and the performed analyses. NSE: "Nash Sutcliffe Efficiency". KGE="Kling Gupta Efficiency". SWAE="Sum of weighted absolute errors", ANOVA="Analysis of variance". Brown et al. (2010) is given as benchmark paper for the verification of ensemble hydrological forecasts.

| *Paper* | Zappa et al. | Addor et al. | Liechti et al. | Antonetti et al. | Antonetti et al. | Antonetti et al. | Horat et a |
|---|---|---|---|---|---|---|---|
| *Year* | 2011 | 2011 | 2013 | 2017 | 2018 | 2018 | 2018 |
| *Journal* | Atm. Research | HESS | HESS | Hydrol. Proc. | HESS | NHESSD | NHESSI |
| **Target areas** | | | | | | | |
| Verzasca | X | | X | | | | X |
| Sihl | | X | | | | | |
| Emme | | | | | X | X | |
| Other | | | X | X | | | |
| **Topics** | | | | | | | |
| Forecasting | X | X | X | | | X | X |
| Model development | | | | X | | | |
| Uncertainty propagation | X | | X | | X | (X) | (X) |
| Intercomparison | | X | X | (X) | (X) | X | X |
| **Model/module** | | | | | | | |
| PREVAH-HRU | X | X | X | | | | X |
| RGM-PRO | | | | X | X | X | X |
| RGM-TRD | | | | | X | X | |
| **Rainfall forcing** | | | | | | | |
| Intrepolated gauges | X | X | X | | X | | X |
| CombiPrecip | | | | X | X | X | X |
| COSMO-1 | | | | | | X | X |
| COSMO-2 | X | X | X | | | | (X) |
| COSMO-LEPS | X | X | | | | | (X) |
| COSMO-E | | | | | | X | X |
| Weather radar nowcasting | X | | X | | | | |
| Frequency | continuous | continuous | events | events | events | events | events |
| Period | 2007-2010 | 2007-2009 | 2007-2010 | 2005-2016 | 2005-2016 | 2016 | 2016 |
| **Analyses** | | | | | | | |
| NSE/KGE | NSE | | | KGE | KGE | NSE/KGE (NSE/KGE) | |
| Brown et al., 2010 | (X) | X | X | | | X | X |
| MonteCarlo | X | | (X) | X | X | X | (X) |
| Other | SWAE | | | | ANOVA | | |

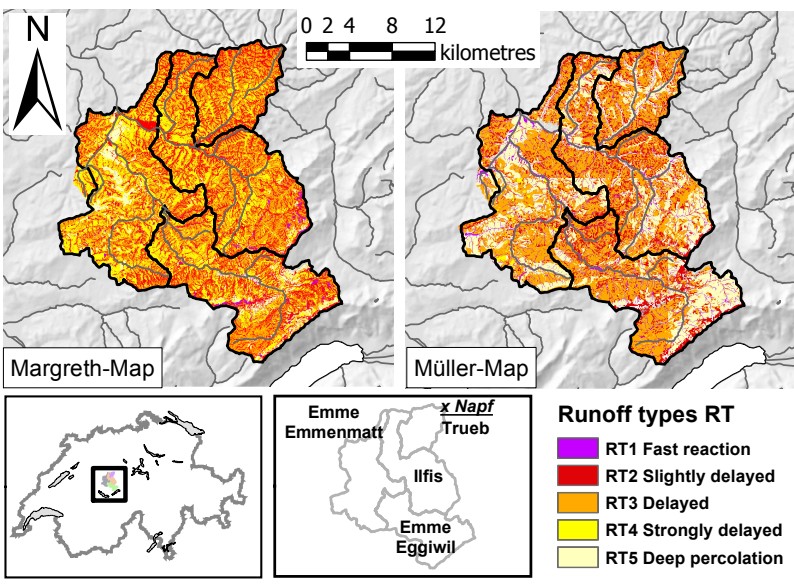

**Figure 1.** Overview on the investigated Emme catchment. Top-left: process map after Margreth et al. (2010); Top-right: process map after Müller et al. (2009); Bottom-left: location of the target area within Switzerland. Bottom-center: situation of the nested sub-areas and location of the Napf meteorological station. GIS elements reproduced by kind authorisation of *swisstopo* (JA022265), BFS GEOSTAT/BUWAL.

**Table 2.** Attribution of DRPs to RTs. Nomenclature of the DRPs: HOF = Hortonian Overland Flow; SOF = Saturation Overland Flow; SSF = Subsurface Flow; DP = Deep Percolation. The suffix *1* is attributed to landscapes showing immediate reaction in terms of runoff generation. The suffixes *2* and *3* mean delayed and strongly delayed reaction, respectively. Sources: Scherrer and Naef (2003) and Antonetti and Zappa (2018). The portion of RT in the maps of Margreth (DRP-ma) and Müller (DRP-mu) is given for the whole target area up to Emmenmatt.

| Runoff Type | Runoff generation | Runoff processes | DRP-ma (%) | DRP-mu (%) |
|:---:|:---:|:---:|:---:|:---:|
| RT1 | Fast | HOF1, HOF2, SOF1 | 2 | 8 |
| RT2 | Slightly delayed | SOF2, SSF1 | 28 | 16 |
| RT3 | Delayed | SSF2 | 36 | 47 |
| RT4 | Strongly delayed | SOF3, SSF3 | 29 | 1 |
| RT5 | Not contributing | DP | 5 | 28 |

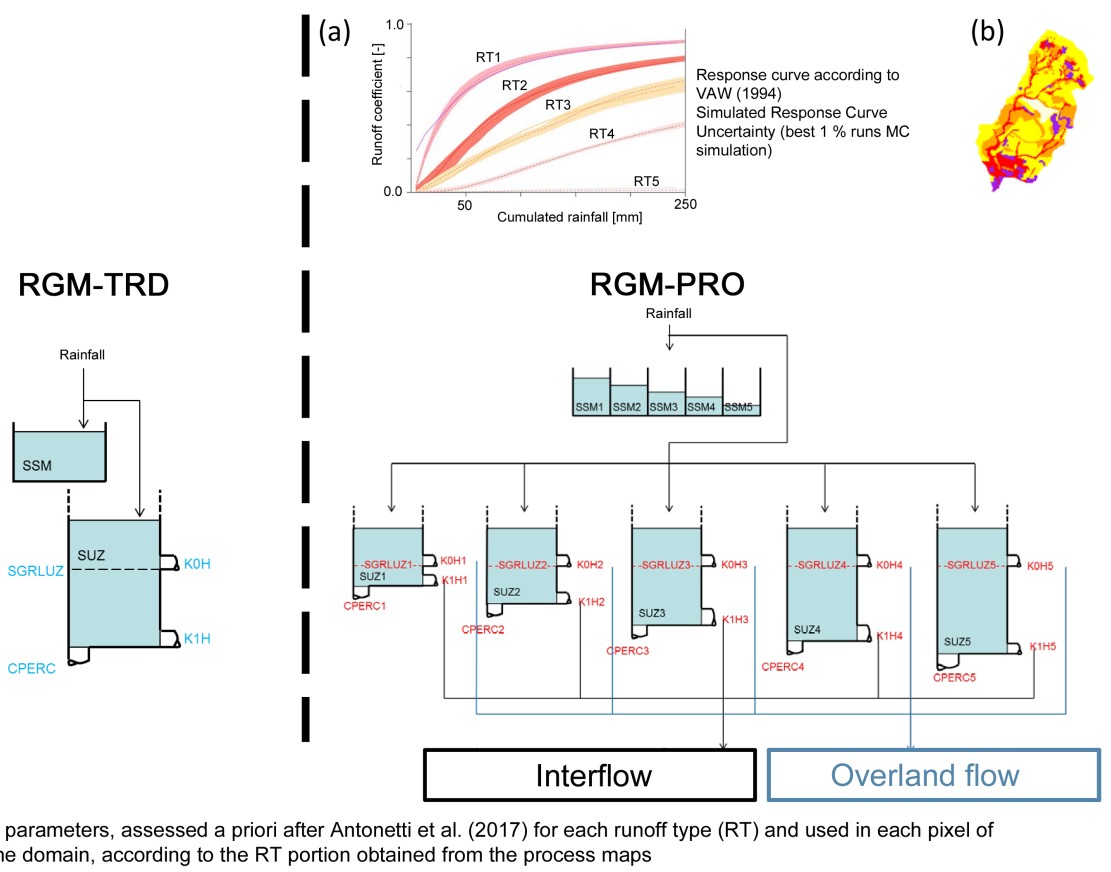

**Figure 2.** Structure of the adopted RGMs. Left: RGM-TRD module from traditional PREVAH with four parameters requiring calibration (Gurtz et al., 2003) and valid for the whole target area. Right: RGM-PRO module using typical infiltration curves (a) for a priori estimation of four parameters for five runoff types (Table 2) after Antonetti et al. (2017) and Antonetti et al. (2016b). RTs are obtained from process maps (b, Fig. 1)

**Table 3.** Description of parameters for RGM-PRO. All parameters have indices for RTs from 1 to 5.

| Abbreviation | Description | Unit |
|---|---|---|
| SGRLUZ1-5 | threshold for overland flow | [mm] |
| K0H1-5 | storage time for overland flow | [h] |
| K1H1-5 | storage time for subsurface flow | [h] |
| CPERC1-5 | maximum percolation rate | [mm/h] |
| SSM1-5 | soil moisture storage | [mm] |
| SUZ1-5 | upper zone runoff storage | [mm] |

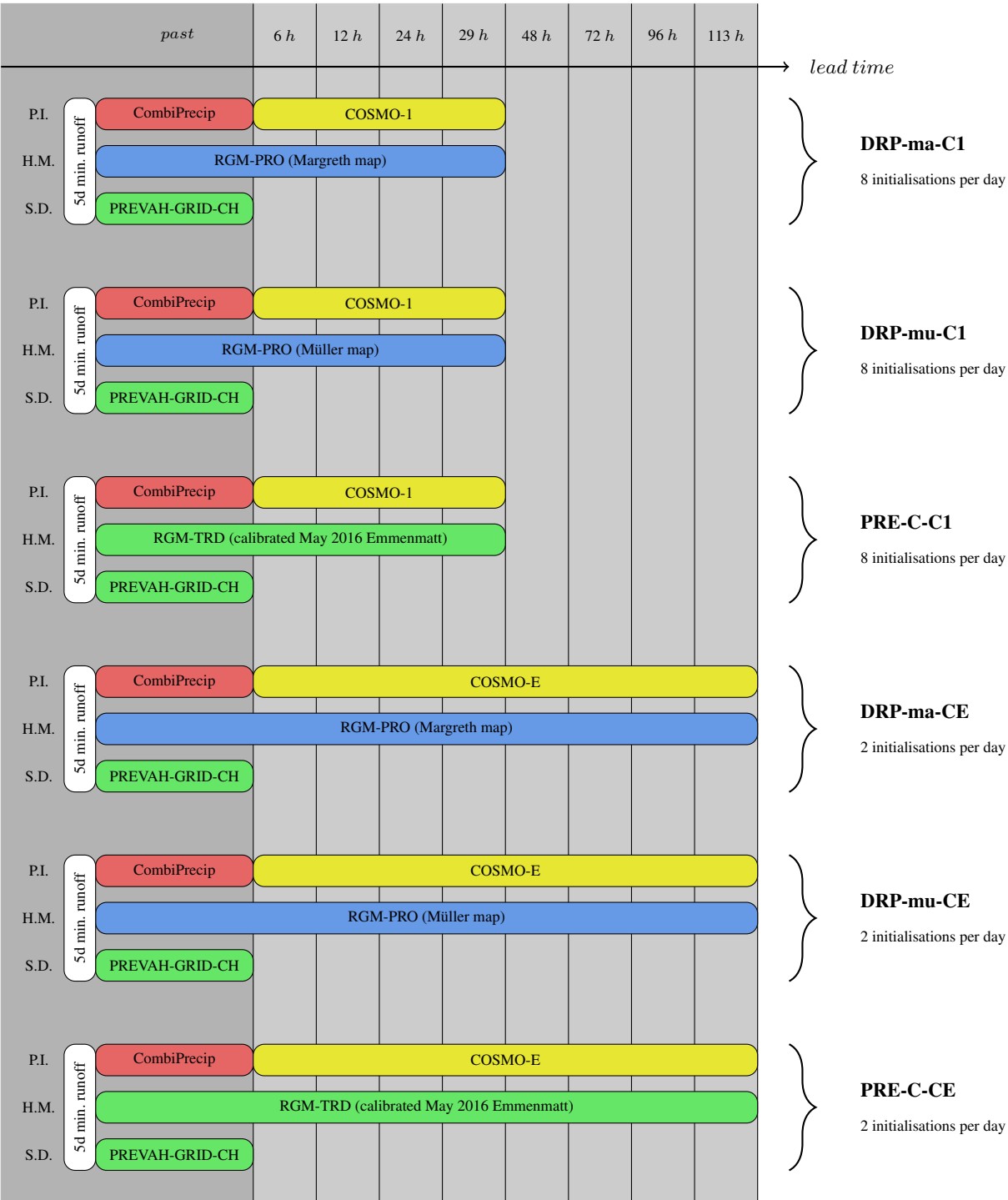

**Figure 3.** Scheme of the FF forecasting chains in the Emme catchments investigated in this paper with keys indicated on the right hand side. On the left hand side, P.I. stands for *precipitation input*, H.M. for *hydrological model* and S.D. for *soil moisture data*.

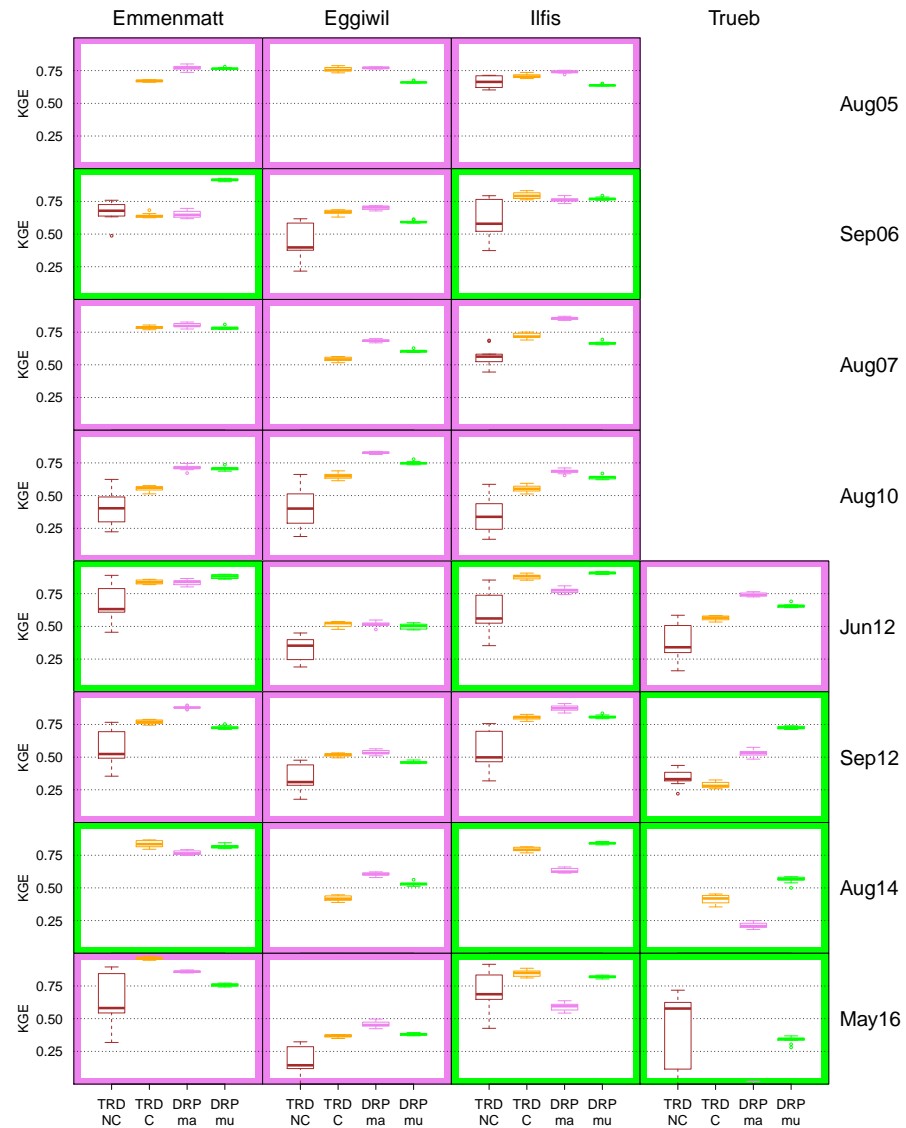

**Figure 4.** Re-analysis of eight large runoff events from 2005 to 2016 with CombiPrecip data in the four Emme catchments. The boxplots represent the simulation results of the different model configurations. *TRD-NC* stands for uncalibrated PREVAH, *TRD-C* for calibrated PREVAH, *DRP-ma* for RGM-PRO relying on Margreth map and *DRP-mu* for RGM-PRO based on Müller map. Border color reveals whether DRP-ma (pink) or DRP-mu (light green) performed better in terms of median KGE. Where no boxplot is visible, all KGE values were below zero. The spread in KGE of uncalibrated PREVAH results from ten random hydrological parameter sets out of 4'000 Monte Carlo simulations, whereas uncertainty from calibrated PREVAH originates from the ten parameter sets with highest KGE for the May16 event in Emmenmatt. Spread in RGM-PRO arises from simulations with ten hydrological parameter sets lying in narrow ranges determined by sprinkler experiments (Antonetti et al., 2017).

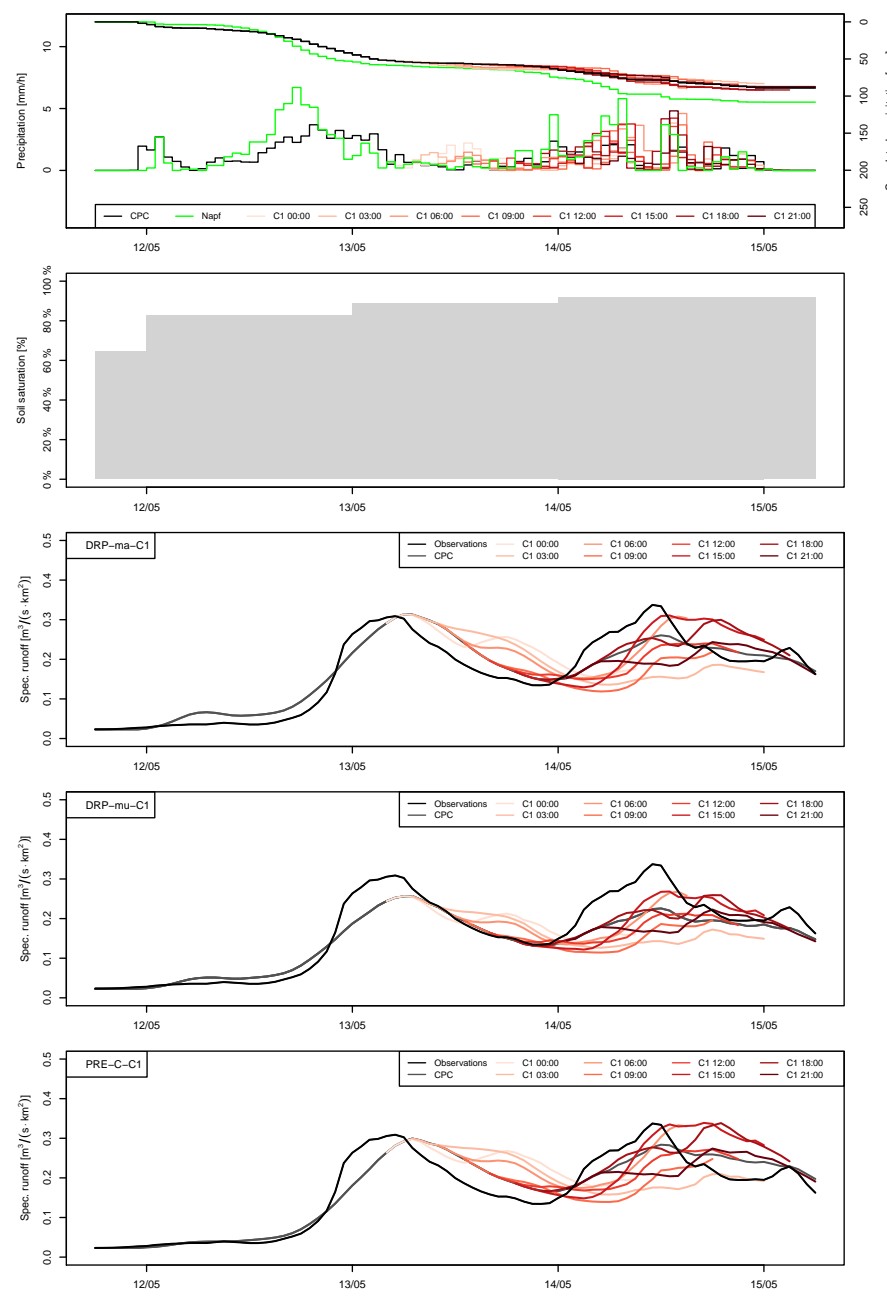

**Figure 5.** Flood predictions with DRP-ma-C1 (middle), DRP-mu-C1 (second lowest) and PRE-C-C1 (lowest panel) for Emmenmatt with eight initialisations each on alert date of 13/5/2016. Catchment precipitation predictions from COSMO-1, measurements from Napf station and CombiPrecip (both provided by MeteoSwiss) are depicted in the uppermost, and evolution of soil moisture from PREVAH simulations in the second panel.

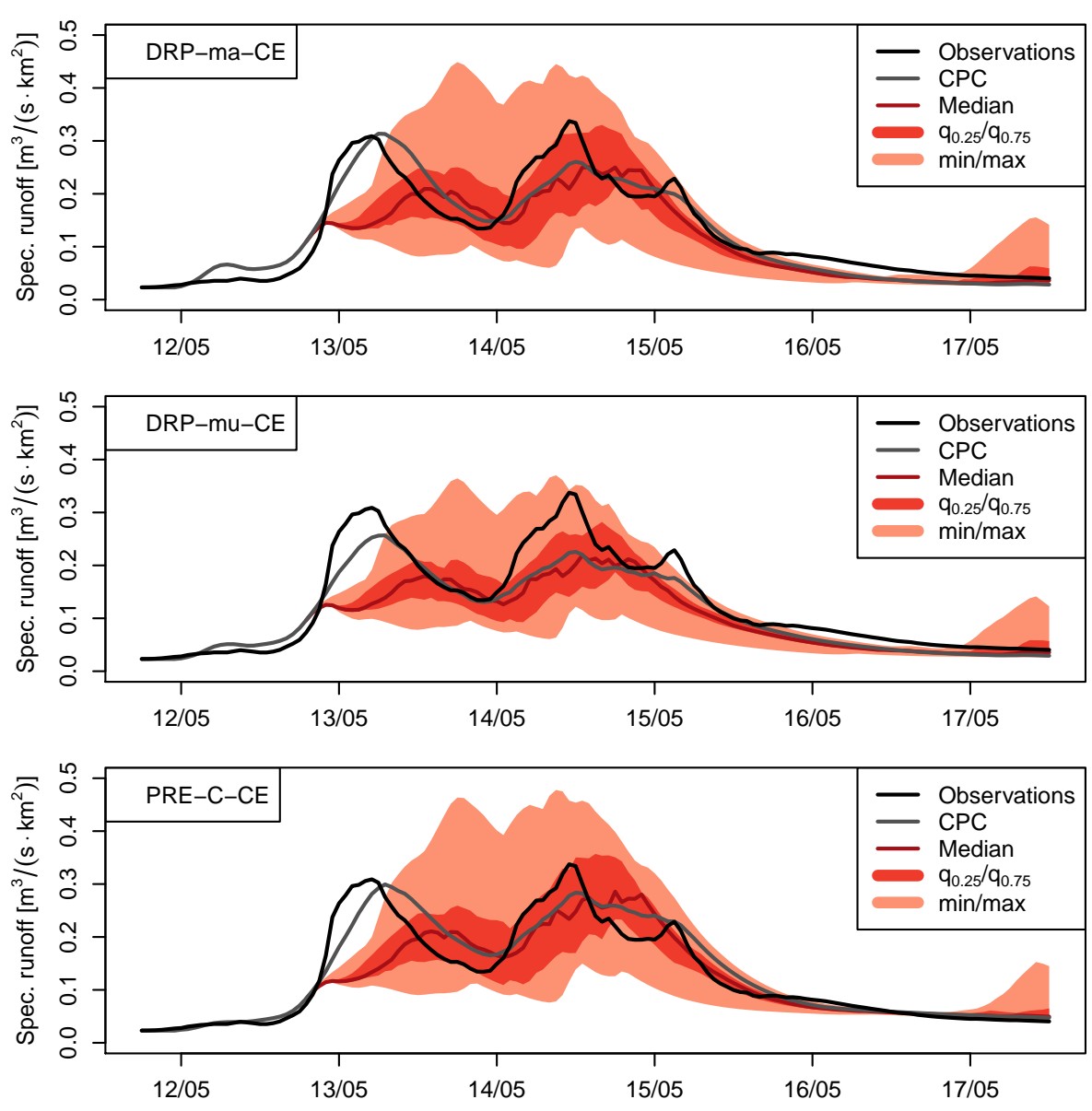

**Figure 6.** Probabilistic forecasts for Emmenmatt catchment with switch to forecast mode on May 12, 19:00 for DRP-ma-CE (top), DRP-mu-CE (middle) and PRE-C-CE (bottom panel).

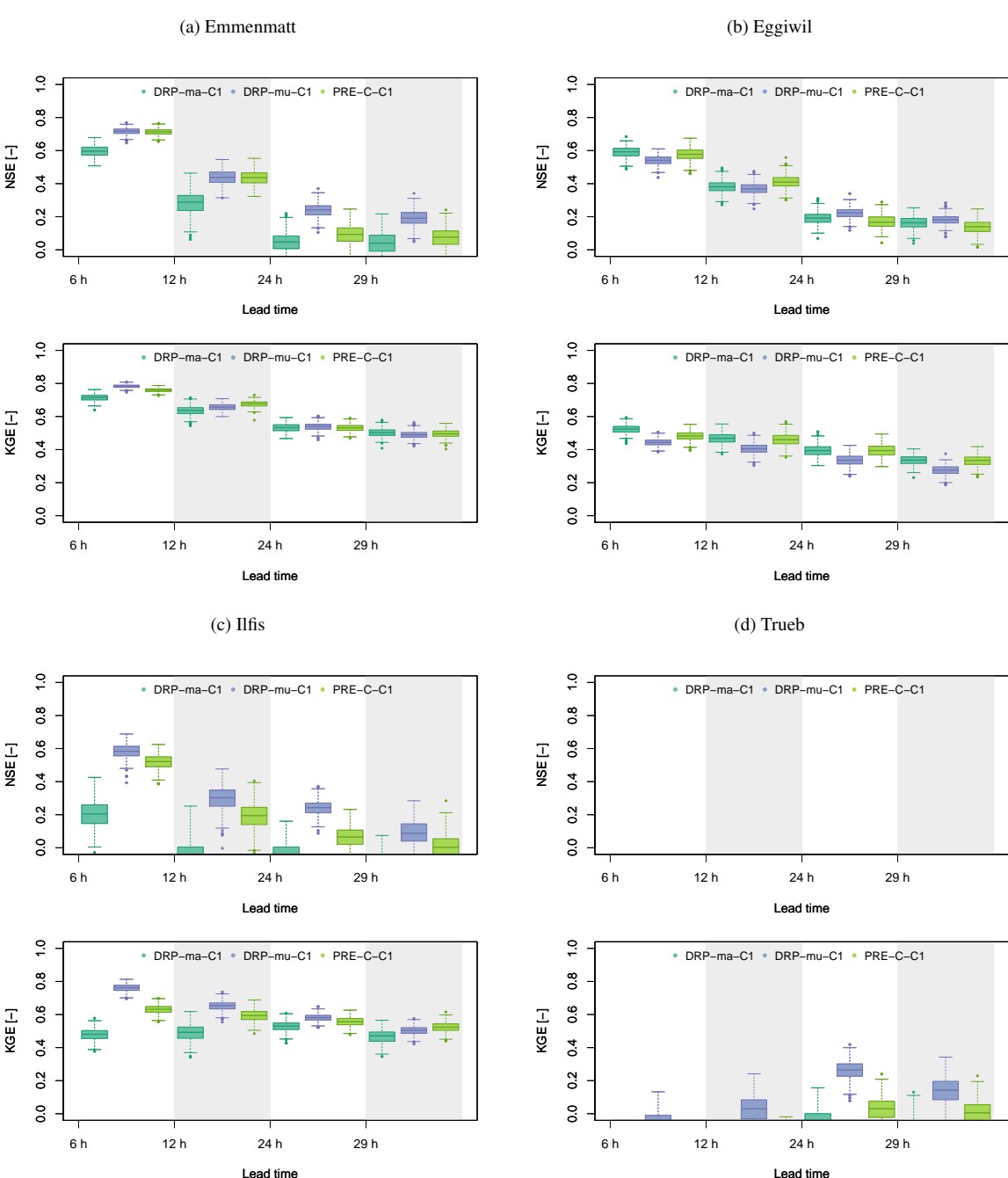

**Figure 7.** NSE and KGE for Emmenmatt, Eggiwil, Ilfis and Trueb as a function of lead time for DRP-ma-C1 (dark green), DRP-mu-C1 (blue) and PRE-C-C1 (light green) for all events in the investigated period from May until July 2016. A window of 6 hours was taken for the computations, e.g. from 19 h to 24 h for the 24 h lead time. The boxplots represent the sampling uncertainties of the score computations obtained with bootstrapping (Addor et al., 2011).

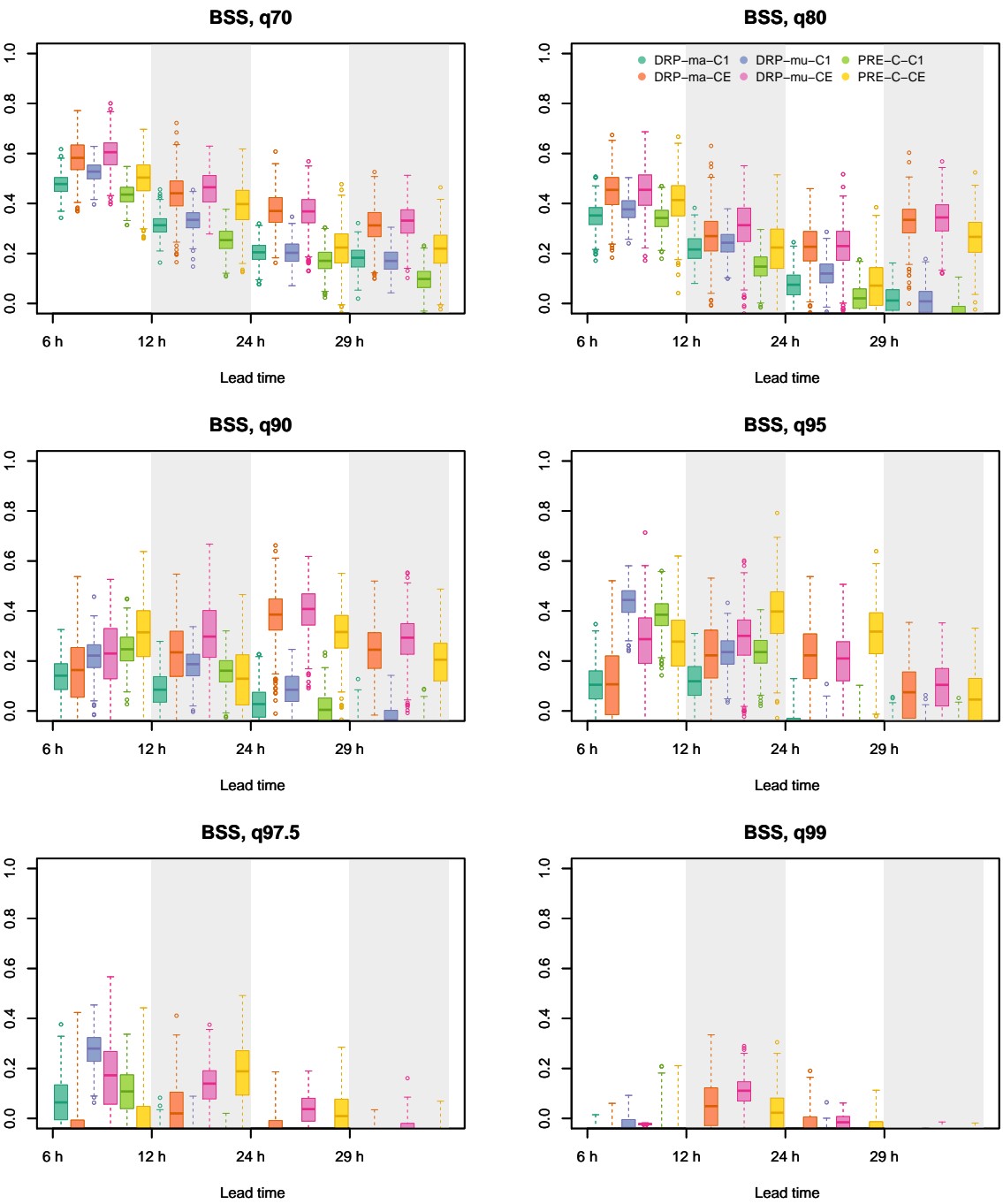

**Figure 8.** Comparison of BSS in Emmenmatt catchment for deterministic DRP-ma-C1, DRP-mu-C1, PRE-C-C1 and probabilistic DRP-ma-CE, DRP-mu-CE, PRE-C-CE as a function of lead time for several threshold quantiles for all events in the investigated period from May until July 2016. A window of 6 hours was taken for the computations. The boxplots represent the sampling uncertainties of the score computations obtained with bootstrapping.

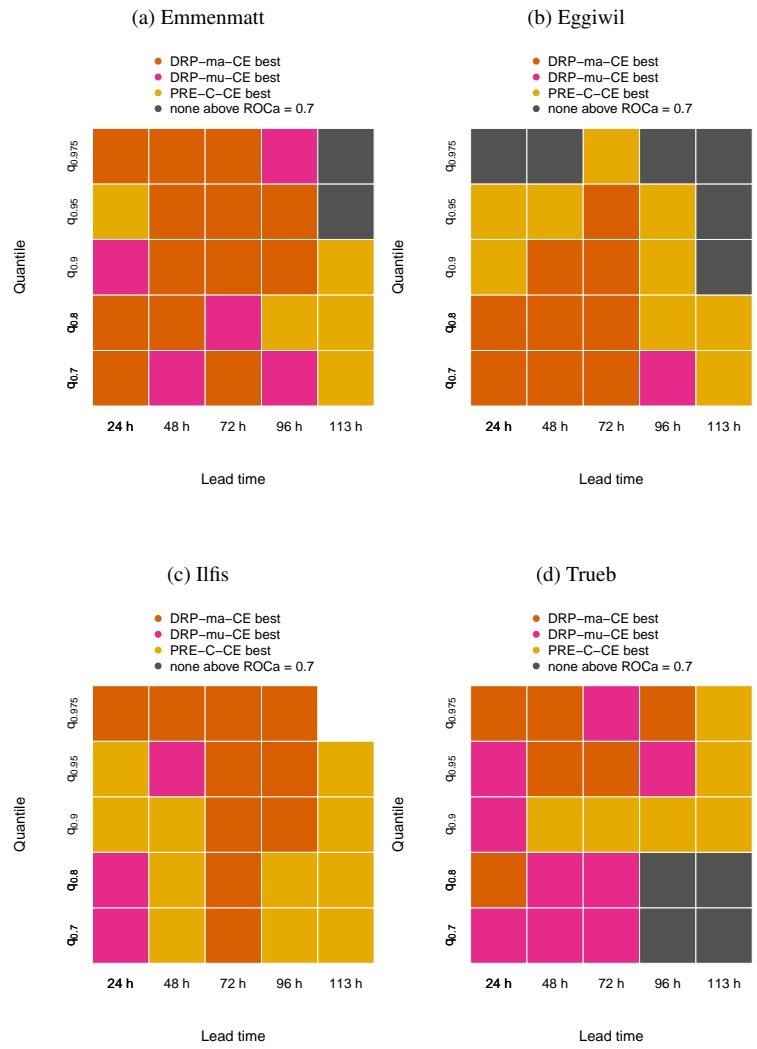

**Figure 9.** Summaries of ROCa for Emmenmatt (a), Eggiwil (b), Ilfis (c) and Trueb (d) as a function of lead time and threshold quantile for DRP-ma-CE, DRP-mu-CE and PRE-C-CE for all events in the investigated period from May until July 2016. Dark orange colour indicates that ROCa of DRP-ma-CE is highest, whereas pink and yellow colour imply that DRP-ma-CE and PRE-C-CE, respectively, perform best. Grey shading indicates that none of the forecasting chains has ROCa higher than 0.7, which is considered to be the minimum value useful for decision makers (Buizza et al., 1999). Please note that there was not enough data for the calculations in Ilfis catchment for the 113 hours lead time and the $q_{0.975}$ threshold quantile.