# Peer review of "Ensemble flood forecasting considering dominant runoff processes: I. Setup and application to nested basins (Emme, Switzerland)"

_Natural Hazards and Earth System Sciences, 2018_

## Referee Comment (RC1) · Anonymous Referee #1 · 4 Jun 2018

GENERAL COMMENTS AND RECOMMENDATION

The manuscript presents a study comparing different setups of the PREVAH model in a nested catchment configuration in the Canton of Berne (Switzerland). Both the topic of the manuscript and the methods used are very interesting. However, I find that the presentation requires significant improvement. Particularly, in the current version the manuscript is somewhat disorganized and does not stand by itself (the authors send the reader to external references in an excessive number of times). This affects the understanding of the methods and the relevancy of the results. Therefore, I think that

the manuscript requires further work before it can be recommended for publication.

MAJOR COMMENTS

1) Introduction: The organization of the first section requires some sharpening and reorganization. Section 1.1 gives too much detail on some of the approaches and very little detail about some others.

2) The presentation of the datasets and methods used in the study should be improved. The authors could focus on the following points:

a. Description of the datasets should include their origin and resolutions. In particular, a brief description of the analyzed COSMO configurations is missing.

b. Calibration of RGM-PRO and RGM-TRD. Given that one of the main differences between the two systems is in the way they are calibrated, the authors should provide a detailed description of the calibration process.

c. How were the uncalibrated and calibrated versions of PREVAH set up? Was the TRD-UC configuration calibrated using the observations of a single event? Did the authors use uniform parameters in the Emme catchment?

d. I find Section 2 too long. The authors could consider splitting it into "Target area and datasets", and "Models and methods."

MINOR COMMENTS

1) Page 4, line 13: "which requires a high model resolution" I guess that the text refers to high NWP model resolution, but it would be worth making it explicit.

2) Page 5, lines 26-29: "For the Trueb catchment, measurements from the Bau-, Verkehr- und Energiedirektion of the Canton of Berne were available. For the evaluation of hindcasts, only four events are investigated as runoff data is not available from 2005 to 2010." It is not fully clear that the last sentence refers to the Trueb catchment.

3) Page 6, lines 10-15: How was CombiPrecip applied in the study? Was the operational CombiPrecip product the one applied here? Were the rain gauge measurements of the Napf station blended in CombiPrecip?

4) Page 7, lines 1-9: It is difficult to follow the retrieval of the RTs maps. Could you provide some details about the Magreth map of SoilCom GmbH, DEM used for the Müller map, resolution...?

5) Page 7, line 31. Why is RGM-PRO event-based? Is this the only alternative?

6) Page 8, lines 4-7: It is surprising that the calibration of the RGM-TRD was done using a single event. How could this affect the performance of the system?

7) Figure 4 - caption: In the figure "Uncalibrated PREVAH" is referred to as "NC".

8) Figure 7 – caption: Please, specify the duration of the analysis. Is this for the event of 12 May? Over what period?

9) Please, refer to the accepted version of the work of Kienzler and Naef (2008): Kienzler, P. M. and Naef, F.: Temporal variability of subsurface stormflow formation, Hydrol. Earth Syst. Sci., 12, 257-265, https://doi.org/10.5194/hess-12-257-2008, 2008.

---

## Referee Comment (RC2) · Anonymous Referee #2 · 2 Jul 2018

**1   Evaluation of principal criteria**

Scientific Significance: Good (2) to Fair (3)

Scientific Quality: Good (2) to excellent (1)

Presentation Quality: Excellent (1)

[Figure]

**2 General comments**

The paper investigates the potential of a process-based runoff generation module, compared to classical (i.e., calibrated) runoff generation assessment with a special focus on smaller, nested, and potentially ungauged catchments.

The paper itself is well-written and well-structured and features high-quality figures. There are only little technical issues, which were found by the reviewer and which are addressed in the respective section ("technical corrections").

However, it should be considered that the paper has some conceptual/methodological limitations:

- The investigation period (of some months) is very limited in order to gain substantial data for further statistical analyses (e.g., skill determination, etc.). However, the authors address and discuss this issue in their text.

- The number of investigated catchments is low; conclusions on regional transferability of the suggested methods remains limited due to lacking statistical significance. However, the authors anticipate these shortcomings within the study design (by using MC methods) and briefly discuss the matter.

For the reviewer, it was a bit hard to get the methodological (?) connection of the reviewed paper to the "accompanying paper" (Horat et. al., 2018), as well as to previous work of the researchers (e.g., Zappa et al., 2011 or Antonetti, 2017). Maybe, a graphical, structured representation of the questions covered in those papers would offer a way to better comprehend the overarching research activities of authors/group/lab and serve the scientific significance of the manuscript.

**3 Specific comments**

P1-L1: Add "potential" before the word "risk" >> if there is no vulnerability, a hazard (i.e., heavy rain) would pose no risk.

P1-L21: What is meant by "satisfying skill" here? Please give a brief comment on that in the manuscript.

P2-L31: Maybe add a reference to Collier & Fox (2003) who propose a Flash Flood Susceptibility Assessment Procedure (FFSAP) which is quite comparable to the one proposed by Mani et. al (2012).

P2-L32: What is meant here by the word "torrent"?

P3-L2: What is determined "with radar" and why?

P4-Ls27-30: The systematic of "physically-based" and "conceptual" models excludes other approaches (e.g., FFSAPs) which are potentially useful for deriving information on "timing and magnitude" (e.g., see http://www.hochwasserzentrum.sachsen.de/fruehwarnung which is based on a simple FFSAP).

P5-L4: Does this mean "assimilation (of transient data)"? If this is not the case, would not it be better to talk of "estimation", rather than "assimilation"?

P5-L12: What is the "statistical approach"? Better replace with something like "skill assessment procedures..."?

P8-L1: What is meant by "... served as fingerprint"?

P8-L3: "Traditional benchmark version..." ... of what?

P9-L2: From our point of view, using continuous measures for skill assessment (e.g., NSE or KGE) should be called "validation". On the other hand, employing event or threshold-based (binary, dichotomous) methods (e.g., AUC or BSS), it is "verification".

P11-L5: Are the probabilistic forecasts post-processed? This should be stated if this applies. . .

**4 Technical corrections**

P5-L8: Please rewrite ". . . model is executed at the runoff gauge" (language).

P8-L8: Better replace "completed" with "conducted".

P19-L21: "bericht" should be "Bericht" (capitalized).

Larger figures/numbers (e.g., "2120") are not separated in the manuscript. Please check, if this is in agreement with the NHESS style guide (e.g., be written as "2,120").

---

## Author Comment (AC1) · 10 Sep 2018

**Ensemble flood forecasting considering dominant runoff processes: I. Setup and application to nested basins (Emme, Switzerland)**

Authors replies to RC1:

We want to thank the reviewer for his/her assessment of our manuscript. We are glad that the topics of our paper have been found to be interesting. In the following we give our answers to the comments and recommendations that have been raised. Reviewer comments RC are **bold,** our reply AR is in *italic*. Insertions in the revised manuscript MI are underlined.
* * *
**RC: I find that the presentation requires significant improvement. Particularly, in the current version the manuscript is somewhat disorganized and does not stand by itself (the authors send the reader to external references in an excessive number of times).**

*AR: The reviewer is right in his statement, that the current paper might not stand by itself in its current organization. As this manuscript is part I of 2 companion papers, we tried to avoid redundancies between the two manuscripts and also to avoid duplicating too large portions of the papers by Antonetti et al. (2016, 2017 and 2018). Reviewer RC2 suggests to include a diagram showing the logic of our past and recent papers in order clarify where relevant information on the adopted data. The diagram will be uploaded in our comment to RC2. This diagram will be also provided as supplementary material to the revised manuscript.*

*In the revised paper we will also replace some of the "external" references to the other manuscripts with short descriptions of the previously referenced topic.*
* * *
**MAJOR COMMENTS**

**RC: 1) Introduction: The organization of the first section requires some sharpening and reorganization. Section 1.1 gives too much detail on some of the approaches and very little detail about some others.**

*AR: In order to streamline section 1.1 and correct the imbalance with the other parts of the introduction we removed a full paragraph (P3 L 25-33 of the original manuscript). Further adjustments have been taken in accordance with the comments of reviewer RC2.*
* * *
**RC: 2) The presentation of the datasets and methods used in the study should be improved. The authors could focus on the following points:**

   **a. Description of the datasets should include their origin and resolutions. In particular,a brief description of the analyzed COSMO configurations is missing.**

**b. Calibration of RGM-PRO and RGM-TRD. Given that one of the main differences between the two systems is in the way they are calibrated, the authors should provide a detailed description of the calibration process.**

**c. How were the uncalibrated and calibrated versions of PREVAH set up? Was the TRD-UC configuration calibrated using the observations of a single event? Did the authors use uniform parameters in the Emme catchment?**

**d. I find Section 2 too long. The authors could consider splitting g it into "Target area and datasets", and "Models and methods."**

*AR: We will accommodate the reviewer request and split the section as suggested (d). When preparing the two manuscripts we decided to put the focus of this first paper on the DRP model and on the description of COMBIPRECIP and present the details on COSMO in the companion paper. Furthermore COSMO data are also introduced as follows:*

*MI: "As future rainfall input, quantitative precipitation forecasts were used from NWP by MeteoSwiss, namely COSMO-E and COSMO-1, and were processed as in Addor et al. (2011). COSMO-1 has a grid spacing of 1.1 km and runs as deterministic model with initialisations every three hours. Lead time is 33 hours except for the 03 UTC run, where a 45 hour forecast is 10 available. COSMO-E is an ensemble prediction system with 2.2 km grid spacing, two initialisations each day and a lead time of 120 hours. Both COSMO-E and COSMO-1 are available for only one season and there is no prior experience in applying these models in a forecasting chain."*

*(b,c) A detailed description of how the two models have been configured (RGM-PRO) and calibrated (RGM-TRD), would basically duplicate the papers Antonetti et al. (2017) and Antonetti et al. (2018, published in the meantime).The reviewer is of the opinion of dealing with a "calibrated" and "uncalibrated" version of our "classic model PREVAH, while as described, RGM-PRO is a new module, configured a priori as presented in Antonetti et al. (2017), where the concepts of runoff generation are strongly oriented to the dominant runoff process approach with 5 separated storage according to the runoff types, while RGM-TRD is identical with the runoff generation module integrated in PREVAH, where one concepts fits all processes (as shown in Figure 2). As far as the calibration of the TRD (one set for the whole basin) version is concerned, yes, only one event is used. We choose this approach in order to have a setup with minimum requirements of observed discharge. This should show the potential of the TRD approach, when a single measurement campaign is available, as discussed for example in Pool et al., (2017).*

*We will adapt the sections on RGM-PRO and RGM-TRD to include these argumentations and, more important, we will better declare RGM-PRO and RGM-TRD as standalone modules.*
* * *
**MINOR COMMENTS**

**RC: 1) Page 4, line 13: "which requires a high model resolution" I guess that the text refers to high NWP model resolution, but it would be worth making it explicit.**

*AR: Will be accommodated*

**RC: 2) Page 5, lines 26-29: "For the Trueb catchment, measurements from the Bau-, Verkehr- und Energiedirektion of the Canton of Berne were available. For the evaluation of hindcasts, only four events are investigated as runoff data is not available from 2005 to 2010." It is not fully clear that the last sentence refers to the Trueb catchment.**

*AR: Will be accommodated*

**RC: 3) Page 6, lines 10-15: How was CombiPrecip applied in the study? Was the operational CombiPrecip product the one applied here? Were the rain gauge measurements of the Napf station blended in CombiPrecip?**

AR: A reanalysis of COMBIPRECIP is available for the period 2005 to 2013. Since 2013 we receive the operational COMBIPRECIP product, which includes the station of Napf in the blending procedure.

**RC: 4) Page 7, lines 1-9: It is difficult to follow the retrieval of the RTs maps. Could you provide some details about the Magreth map of SoilCom GmbH, DEM used for the Müller map, resolution. . .?**

*AR: The RT maps used coincides with the products developed in the Antonetti and Zappa (2018) study. As detailed in Antonetti et al. (2016) the "SoilCom" map requires more input information than the Müller Map and uses a DTM in 2 m resolution. The Müller map is obtained from a DTM of 25 m resolution. The information required by the reviewer will be added in the revised manuscript.*

**RC: 5) Page 7, line 31. Why is RGM-PRO event-based? Is this the only alternative?**

*AR: We did not want to develop a new fully fledged model, but develop something that we can setup ad hoc in any area and feed with precipitation information only, while prescribing initial soil moisture deficit, in this case from a nation wide simulation of the "donor" model PREVAH. This tool should be only active when thunderstorms are to be expected and provide information to anticipate flash-floods in fast-reacting area small areas, while for larger catchments the "established" PREVAH versions are adequate. We will integrate some of these thoughts when we first introduce RGM-PRO.*

**RC: 6) Page 8, lines 4-7: It is surprising that the calibration of the RGM-TRD was done using a single event. How could this affect the performance of the system?**

*AR: We replied to the second question above, i.e. to mimic nearly ungauged conditions. For sure calibration over several events would have led to more robust parameters.*

**7) Figure 4 - caption: In the figure "Uncalibrated PREVAH" is referred to as "NC".**

*AR: corrected, thanks*

**8) Figure 7 – caption: Please, specify the duration of the analysis. Is this for the event of 12 May? Over what period?**

AR: It refers to all events considered events between May and July. Caption will be modified accordingly.

**AR: We assume the reviewer refers to Figure 6. The analyses cover the**

**9) Please, refer to the accepted version of the work of Kienzler and Naef (2008): Kienzler, P. M. and Naef, F.: Temporal variability of subsurface stormflow formation, Hydrol. Earth Syst. Sci., 12, 257-265, https://doi.org/10.5194/hess-12-257-2008, 2008.**

*AR: corrected, thanks*

References:

Antonetti, M., Buss, R., Scherrer, S., Margreth, M., and Zappa, M.: Mapping dominant runoff processes: an evaluation of different approaches using similarity measures and synthetic runoff simulations, Hydrol. Earth Syst. Sci., 20, 2929-2945, https://doi.org/10.5194/hess-20-2929-2016, 2016.

Antonetti, M., Scherrer, S., Kienzler, P. M., Margreth, M., & Zappa, M. (2017). Process-based hydrological modelling: the potential of a bottom-up approach for runoff predictions in ungauged catchments. *Hydrological Processes*, *31*(16), 2902-2920. https://doi.org/10.1002/hyp.11232

Antonetti, M., & Zappa, M. (2018). How can expert knowledge increase the realism of conceptual hydrological models? A case study based on the concept of dominant runoff process in the Swiss Pre-Alps. *Hydrology and Earth System Sciences*, *22*(8), 4425-4447. https://doi.org/10.5194/hess-22-4425-2018

Pool, Sandra; Viviroli, Daniel; Seibert, Jan (2017). Prediction of hydrographs and flow-duration curves in almost ungauged catchments: Which runoff measurements are most informative for model calibration? Journal of Hydrology, 554:613-622.

---

## Author Comment (AC2) · 11 Sep 2018

**Ensemble flood forecasting considering dominant runoff processes: I. Setup and application to nested basins (Emme, Switzerland)**

Authors replies to RC2:

We want to thank the reviewer for his/her assessment of our manuscript. In the following we give our answers to the comments and recommendations that have been raised. Reviewer comments RC are **bold,** our reply AR is in *italic*. Insertions in the revised manuscript MI are underlined.
* * *
**MAJOR COMMENTS**

**RC: However, it should be considered that the paper has some conceptual/methodological limitations:**

- **The investigation period (of some months) is very limited in order to gain substantial data for further statistical analyses (e.g., skill determination, etc.). However, the authors address and discuss this issue in their text.**
- **The number of investigated catchments is low; conclusions on regional transferability of the suggested methods remains limited due to lacking statistical significance. However, the authors anticipate these shortcomings within the study design (by using MC methods) and briefly discuss the matter.**

*AR: We are of course aware, that more basins and longer periods of evaluation are always welcome. NHESS is in this respect a journal that regularly publishes case studies (e.g. Kobayashi et al., 2016; Cane et al., 2013), preliminary assessments (Picciotti et al., 2013) or intercomparison of approaches during limited period of time (e.g. Davoli et al., 2018). Having targeted NHESS as journal for disseminating our experience, this study is designed to evaluate two different model structures, during a representative flood season and in case of nested basins with different area. With this approach we can learn about the quality of the novel approaches from different perspectives at the same time. The transfer of experience to another catchment and climatic region is presented in the companion paper by Horat et al. (2018). As far as the length of the investigation period is concerned, some limitations arise from the use of COSMO-E and COSMO-1. MeteoSwiss decommissioned after several years the antecedent operational NWP COSMO-2 and COSMO-LEPS in 2016. As we want to make our systems operational, it was for us important to focus on a first analysis with the NEW NWPS that we receive and archive in real-time since February 2016.*

*In the revised manuscript we will better declare our choices concerning selection of basins and investigation period..*

**RC: For the reviewer, it was a bit hard to get the methodological (?) connection of the reviewed paper to the "accompanying paper" (Horat et. al., 2018), as well as to previous work of the researchers (e.g., Zappa et al., 2011 or Antonetti, 2017). Maybe, a graphical, structured representation of the questions covered in those papers would offer a way to better comprehend the overarching research activities of authors/group/lab and serve the scientific significance of the manuscript.**

*AR: With thank the reviewer for this suggestion. We first taught to follow his comment and create an image, but finally opted for a table that we will insert in the supplementary material of the revised manuscript. The table is also provided here below.*

| Paper | Zappa et al. | Addor et al. | Liechti et al. | Antonetti et al. | Antonetti et al. | Antonetti et al. | Horat et al. |
|---|---|---|---|---|---|---|---|
| Year | 2011 | 2011 | 2013 | 2017 | 2018 | 2018 | 2018 |
| Journal | At. Research | HESS | HESS | Hydrol. Proc. | HESS | NHESSD | NHESSD |
| **Target areas** | | | | | | | |
| Verzasca | X | | X | | | | X |
| Sihl | | X | | | | | |
| Emme | | | | | X | X | |
| Other | | | X | X | | | |
| **Topics** | | | | | | | |
| Forecasting | X | X | X | | | X | X |
| Model development | | | | X | | | |
| Uncertainty propagation | X | | X | | X | (X) | (X) |
| Intercomparison | | X | X | (X) | (X) | X | X |
| **Model/module** | | | | | | | |
| PREVAH-HRU | X | X | X | | | | X |
| RGM-PRO | | | | X | X | X | X |
| RGM-TRD | | | | | X | X | |
| **Rainfall forcing** | | | | | | | |
| Intrepolated gauges | X | X | X | | X | | X |
| Combiprecip | | | | X | X | X | X |
| COSMO-1 | | | | | | | |
| COSMO-2 | X | X | X | | | | |
| COSMO-LEPS | X | X | | | | X | X |
| COSMO-E | | | | | | X | X |
| Weather radar nowcasting | X | | X | | | | |
| Frequency | continuous | continuous | events | events | events | events | events |
| Period | 2007-2010 | 2007-2009 | 2007-2010 | 2005-2016 | 2005-2016 | 2016 | 2016 |
| **Analyses** | | | | | | | |
| NSE/KGE | NSE | | | KGE | KGE | NSE/KGE | |
| Brier/ROC/FAR/RankHist | (X) | X | X | | | X | X |
| MonteCarlo | X | | (X) | X | X | X | (X) |
| Other | SWAE | | | | ANOVA | | |

*Zappa et al. (2011) is our benchmark paper on uncertainty propagation*

*Addor et al.. (2011) is our reference work on verification of deterministic and ensemble forecasts*

*Liechti et al. (2013) focuses on flash-flood nowcasting with advanced weather radar products*

*Antonetti et al. (2017) introduces RGM-PRO*

*Antonetti et al. (2018, HESS) evaluate structures and configurations of RGM-PRO in the Emme catchment*

*Antonetti et al. (2018, NHESSD) first apply RGM-PRO in forecasting mode for the Emme catchment and is our first study with COSMO-E/COSMO-1*

*Horat et al. (2018, NHESSD) applies RGM-PRO in forecasting mode for the Verzasca catchment and compare its quality with our current operational model as forced by COSMO-E/COSMO-1.*
* * *
**SPECIFIC COMMENTS**

**RC: P1-L1: Add "potential" before the word "risk" >> if there is no vulnerability, a hazard (i.e., heavy rain) would pose no risk.**

*AR: Will be addressed*

**RC: P1-L21: What is meant by "satisfying skill" here? Please give a brief comment on that in the manuscript.**

*AR: Will be addressed*

**RC: P2-L31: Maybe add a reference to Collier & Fox (2003) who propose a Flash Flood Susceptibility Assessment Procedure (FFSAP) which is quite comparable to the one proposed by Mani et. al (2012).**

*AR: We will elaborate on FFSAP in the revised manuscript*

**RC: P2-L32: What is meant here by the word "torrent"?**

*AR: We replaced "torrent" with "flash floods", it was originally meant "torrential flow"*

**RC: P3-L2: What is determined "with radar" and why?**

*AR: We will specify the meaning of "radar", i.e. identification of thresholds according to data from rainfall radar.*

**RC: P4-Ls27-30: The systematic of "physically-based" and "conceptual" models excludes other approaches (e.g., FFSAPs) which are potentially useful for deriving information on "timing and magnitude" (e.g., see http://www.hochwasserzentrum.sachsen.de/fruehwarnung which is based on a simple FFSAP).**

*AR: As already states before we will elaborate on FFSAP in the revised manuscript and also present the application for Saxony (Philipp et al., 2016)*

**RC: P5-L4: Does this mean "assimilation (of transient data)"? If this is not the case, would not it be better to talk of "estimation", rather than "assimilation"?**

*AR: We will rephrase the sentence: "To what extent does the skill of the FF prediction depend on the use of model structures considering spatially distributed information on runoff processes into a hydrological model?"*

**RC: P5-L12: What is the "statistical approach"? Better replace with something like "skill assessment procedures. . . "?**

*AR: Will be addressed*

**RC: P8-L1: What is meant by ". . . served as fingerprint"?**

*AR: We used here the terminology introduced by Blöschl et al. (2009). The sentence will be expanded: "With this method, the map of RTs serves as fingerprint since it contains information determining the spatial variability of soil moisture Antonetti (2018)."*

**RC: P8-L3: "Traditional benchmark version. . . " . . . of what?**

*AR: "Traditional benchmark version with conventional hydrological runoff generation module"*

**RC: P9-L2: From our point of view, using continuous measures for skill assessment (e.g., NSE or KGE) should be called "validation". On the other hand, employing event or threshold-based (binary, dichotomous) methods (e.g., AUC or BSS), it is "verification".**

*AR: We will re-arrange the section, remove the sub-section title and introduce the definition of BSS in order to make the manuscript less dependent on the companion paper.*

**RC: P11-L5: Are the probabilistic forecasts post-processed? This should be stated if this applies.**

*AR: no, they are not.*
* * *
*Technical corrections*

**RC: P5-L8: Please rewrite ". . . model is executed at the runoff gauge" (language).**

*AR: Will be addressed*

**RC: P8-L8: Better replace "completed" with "conducted".**

*AR: Will be addressed*

**RC: P19-L21: "bericht" should be "Bericht" (capitalized).**

*AR: Will be addressed*

**RC: Larger figures/numbers (e.g., "2120") are not separated in the manuscript. Please check, if this is in agreement with the NHESS style guide (e.g., be written as "2,120").**

*AR: Will be addressed*

References:

Addor, N., Jaun, S., Fundel, F., and Zappa, M.: An operational hydrological ensemble prediction system for the city of Zurich (Switzerland): Skill, case studies and scenarios, Hydrology and Earth System Sciences, 15, 2327–2347, doi:10.5194/hess-15-2327-2011, 2011.

Antonetti, M., & Zappa, M. (2018). How can expert knowledge increase the realism of conceptual hydrological models? A case study based on the concept of dominant runoff process in the Swiss Pre-Alps. *Hydrology and Earth System Sciences*, *22*(8), 4425-4447. https://doi.org/10.5194/hess-22-4425-2018

Antonetti, M., Scherrer, S., Kienzler, P. M., Margreth, M., & Zappa, M. (2017). Process-based hydrological modelling: the potential of a bottom-up approach for runoff predictions in ungauged catchments. *Hydrological Processes*, *31*(16), 2902-2920. https://doi.org/10.1002/hyp.11232

Blöschl, G., Reszler, C., and Komma, J.: A spatially distributed flash flood forecasting model, Environmental Modelling and Software, 23, 464–478, doi:10.1016/j.envsoft.2007.06.010, 2008.

Cane, D., Ghigo, S., Rabuffetti, D., and Milelli, M.: Real-time flood forecasting coupling different postprocessing techniques of precipitation forecast ensembles with a distributed hydrological model. The case study of may 2008 flood in western Piemonte, Italy, Nat. Hazards Earth Syst. Sci., 13, 211-220, https://doi.org/10.5194/nhess-13-211-2013, 2013.

Devoli, G., Tiranti, D., Cremonini, R., Sund, M., and Boje, S.: Comparison of landslide forecasting services in Piedmont (Italy) and Norway, illustrated by events in late spring 2013, Nat. Hazards Earth Syst. Sci., 18, 1351-1372, https://doi.org/10.5194/nhess-18-1351-2018, 2018.

Horat, C., Antonetti, M., Liechti, K., Kaufmann, P., and Zappa, M.: Ensemble flood forecasting considering dominant runoff processes: II. Benchmark against a state-of-the-art model-chain (Verzasca, Switzerland), Nat. Hazards Earth Syst. Sci. Discuss., https://doi.org/10.5194/nhess-2018-119, in review, 2018.

Kobayashi, K., Otsuka, S., Apip, and Saito, K.: Ensemble flood simulation for a small dam catchment in Japan using 10 and 2 km resolution nonhydrostatic model rainfalls, Nat. Hazards Earth Syst. Sci., 16, 1821-1839, https://doi.org/10.5194/nhess-16-1821-2016, 2016.

Liechti, K., Panziera, L., Germann, U., and Zappa, M.: The potential of radar-based ensemble forecasts for flash-flood early warning in the southern Swiss Alps, Hydrology and Earth System Sciences, 17, 3853–3869, doi:doi:10.5194/hess-17-3853-2013, 2013.

Picciotti, E., Marzano, F. S., Anagnostou, E. N., Kalogiros, J., Fessas, Y., Volpi, A., Cazac, V., Pace, R., Cinque, G., Bernardini, L., De Sanctis, K., Di Fabio, S., Montopoli, M., Anagnostou, M. N., Telleschi, A., Dimitriou, E., and Stella, J.: Coupling X-band dual-polarized mini-radars and hydro-meteorological forecast models: the HYDRORAD project, Nat. Hazards Earth Syst. Sci., 13, 1229-1241, https://doi.org/10.5194/nhess-13-1229-2013, 2013.

Philipp, A., Kerl, F., Büttner, U., Metzkes, C., Singer, T., Wagner, M., and Schütze, N.: Small-scale (flash) flood early warning in the light of operational requirements: opportunities and limits with regard to user demands, driving data, and hydrologic modeling techniques, Proc. IAHS, 373, 201-208, https://doi.org/10.5194/piahs-373-201-2016, 2016.

Zappa, M., Jaun, S., Germann, U., Walser, A., and Fundel, F.: Superposition of three sources of uncertainties in operational flood forecasting chains, Atmospheric Research, 100, 246–262, doi:doi:10.1016/j.atmosres.2010.12.005, 2011.

---

## Author Response (AR2)

**Ensemble flood forecasting considering dominant runoff processes: I. Setup and application to nested basins (Emme, Switzerland)**

Authors replies to the correction requested by the reviewers and the associate editor.

**Abstract: It would be desirable to avoid abbreviations in the abstract (e.g. FF, RTs, ...)**

Abbreviations have been removed from the abstract

**Page 5, lines 26 - 29 could be moved before the sentence 'The results are shown in Sect. 4 and ...' (line 24)**

Done

Page 7, headline of section 3 'Models Methods' -> 'Models and Methods' ? Done

Page 8, lines 20 and 21: "25 by 25 m" and "500x500 m". "m" should be "m2". Done

**Page 8, 29: calibration of the RGM-TRD based on ne single event. A small justification beyond the provided reference could be informative for the reader**

The justification going beyond the reference is presented two lines before.

"In this application we decided to calibrate on one single event, the largest runoff event measured at Emmenmatt gauge in 2016 which occurred on the 14th of May. We choose this approach in order to evaluate a setup with minimum requirements concerning observed discharge."

Page 11, line 8 (and beyond): In Fig. 4, the configuration "TRD-UC" is referred to as "TRD-NC" We now use TRD-NC consistently in the manuscript.

**Page 11, lines 9-11. "A calibration was completed for the Emmenmatt gauge". Do you refer to the calibration of the TRD-C configuration described in page 8, line 29?**

A reference to section 3.1.3 has been introduced.

**Page 13, lines 4-6. Could the good performance of deterministic forecasting chains at short lead times be explained by the higher resolution of the COSMO-1 grid? In this discussion, the differences between the configurations of COSMO-1 and COSMO-E could be relevant.**

Following sentence has been introduced

"One of the reasons for this behaviour could be the higher resolution of COSMO-1 as compared to COSMO-E and thus the better consideration of convection in the deterministic apporoach. Nevertheless the analysed data set is too short for conclusive statements on this finding."

Best regards on behalf of all co-authors. Massimiliano Zappa

Summary 10.12.2018 12:31:57

**Differences exist between documents.**

| New Document:                      | Old Document:                 |
|------------------------------------|-------------------------------|
| drp_nhess_ant_et_al_revised | drp_nhess_ant_et_al_revised_f |
| 31 pages (2.54 MB)                 | 31 pages (2.54 MB)            |
| 10.12.2018 12:31:45                | 10.12.2018 12:31:45           |
| Used to display results.           |                               |

Get started: first change is on page 1.

No pages were deleted

**How to read this report**

Highlight indicates a change.
Deleted indicates deleted content.
indicates pages were changed.
indicates pages were moved.

[revised manuscript text omitted]
             |              | Х              |                  |                  |                   | Х          |
| Sihl                     |               | Х            |                |                  |                  |                   |            |
| Emme                     |               |              |                |                  | Х                | Х                 |            |
| Other                    |               |              | Х              | Х                |                  |                   |            |
| Topics                   |               |              |                |                  |                  |                   |            |
| Forecasting              | x             | Х            | Х              |                  |                  | Х                 | Х          |
| Model development        |               |              |                | Х                |                  |                   |            |
| Uncertainty propagation  | X             |              | Х              |                  | Х                | (X)               | (X)        |
| Intercomparison          |               | Х            | Х              | (X)              | (X)              | Х                 | Х          |
| Model/module             |               |              |                |                  |                  |                   |            |
| PREVAH-HRU               | X             | Х            | Х              |                  |                  |                   | Х          |
| RGM-PRO                  |               |              |                | Х                | Х                | Х                 | Х          |
| RGM-TRD                  |               |              |                |                  | Х                | Х                 |            |
| Rainfall forcing         |               |              |                |                  |                  |                   |            |
| Intrepolated gauges      | X             | Х            | Х              |                  | Х                |                   | Х          |
| CombiPrecip              |               |              |                | Х                | Х                | Х                 | Х          |
| COSMO-1                  |               |              |                |                  |                  | Х                 | Х          |
| COSMO-2                  | X             | Х            | Х              |                  |                  |                   | (X)        |
| COSMO-LEPS               | X             | Х            |                |                  |                  |                   | (X)        |
| COSMO-E                  |               |              |                |                  |                  | Х                 | Х          |
| Weather radar nowcasting | X             |              | Х              |                  |                  |                   |            |
| Frequency                | continuous    | continuous   | events         | events           | events           | events            | events     |
| Period                   | 2007-2010     | 2007-2009    | 2007-2010      | 2005-2016        | 2005-2016        | 2016              | 2016       |
| Analyses                 |               |              |                |                  |                  |                   |            |
| NSE/KGE                  | NSE           |              |                | KGE              | KGE              | NSE/KGE (NSE/KGE) |            |
| Brown et al., 2010       | (X)           | Х            | Х              |                  |                  | Х                 | Х          |
| MonteCarlo               | X             |              | (X)            | Х                | Х                | Х                 | (X)        |
| Other                    | SWAE          |              |                |                  | ANOVA            |                   |            |